# The Unreasonable Effectiveness of Structured Random Orthogonal Embeddings

**Krzysztof Choromanski** *
Google Brain Robotics
kchoro@google.com

**Mark Rowland** *
University of Cambridge
mr504@cam.ac.uk

**Adrian Weller**
University of Cambridge and Alan Turing Institute
aw665@cam.ac.uk

## Abstract

We examine a class of embeddings based on structured random matrices with orthogonal rows which can be applied in many machine learning applications including dimensionality reduction and kernel approximation. For both the Johnson-Lindenstrauss transform and the angular kernel, we show that we can select matrices yielding guaranteed improved performance in accuracy and/or speed compared to earlier methods. We introduce matrices with complex entries which give significant further accuracy improvement. We provide geometric and Markov chain-based perspectives to help understand the benefits, and empirical results which suggest that the approach is helpful in a wider range of applications.

## 1 Introduction

Embedding methods play a central role in many machine learning applications by projecting feature vectors into a new space (often nonlinearly), allowing the original task to be solved more efficiently. The new space might have more or fewer dimensions depending on the goal. Applications include the Johnson-Lindenstrauss Transform for dimensionality reduction (JLT, Johnson and Lindenstrauss, 1984) and kernel methods with random feature maps (Rahimi and Recht, 2007). The embedding can be costly hence many fast methods have been developed, see §1.1 for background and related work.

We present a general class of random embeddings based on particular structured random matrices with orthogonal rows, which we call *random ortho-matrices* (ROMs); see §2. We show that ROMs may be used for the applications above, in each case demonstrating improvements over previous methods in statistical accuracy (measured by mean squared error, MSE), in computational efficiency (while providing similar accuracy), or both. We highlight the following contributions:

- In §3: The *Orthogonal Johnson-Lindenstrauss Transform* (OJLT) for dimensionality reduction. We prove this has strictly smaller MSE than the previous unstructured JLT mechanisms. Further, OJLT is as fast as the fastest previous JLT variants (which are structured).

- In §4: Estimators for the *angular kernel* (Sidorov et al., 2014) which guarantee better MSE. The *angular kernel* is important for many applications, including natural language processing (Sidorov et al., 2014), image analysis (Jégou et al., 2011), speaker representations (Schmidt et al., 2014) and tf-idf data sets (Sundaram et al., 2013).

- In §5: Two perspectives on the effectiveness of ROMs to help build intuitive understanding.

In §6 we provide empirical results which support our analysis, and show that ROMs are effective for a still broader set of applications. Full details and proofs of all results are in the Appendix.

---

## 1.1 Background and related work

Our ROMs can have two forms (see §2 for details): (i) a $\mathbf{G}_{\text{ort}}$ is a random Gaussian matrix conditioned on rows being orthogonal; or (ii) an **SD**-*product* matrix is formed by multiplying some number $k$ of **SD** blocks, each of which is highly structured, typically leading to fast computation of products. Here **S** is a particular structured matrix, and **D** is a random diagonal matrix; see §2 for full details. Our **SD** block generalizes an **HD** block, where **H** is a *Hadamard* matrix, which received previous attention. Earlier approaches to embeddings have explored using various structured matrices, including particular versions of one or other of our two forms, though in different contexts.

For dimensionality reduction, Ailon and Chazelle (2006) used a single **HD** block as a way to spread out the mass of a vector over all dimensions before applying a sparse Gaussian matrix. Choromanski and Sindhwani (2016) also used just one **HD** block as part of a larger structure. Bojarski et al. (2017) discussed using $k = 3$ **HD** blocks for locality-sensitive hashing methods but gave no concrete results for their application to dimensionality reduction or kernel approximation. All these works, and other earlier approaches (Hinrichs and Vybíral, 2011; Vybíral, 2011; Zhang and Cheng, 2013; Le et al., 2013; Choromanska et al., 2016), provided computational benefits by using structured matrices with less randomness than unstructured iid Gaussian matrices, but none demonstrated accuracy gains.

Yu et al. (2016) were the first to show that $\mathbf{G}_{\text{ort}}$-type matrices can yield improved accuracy, but their theoretical result applies only asymptotically for many dimensions, only for the Gaussian kernel and for just one specific orthogonal transformation, which is one instance of the larger class we consider. Their theoretical result does not yield computational benefits. Yu et al. (2016) did explore using a number $k$ of **HD** blocks empirically, observing good computational and statistical performance for $k = 3$, but without any theoretical accuracy guarantees. It was left as an open question why matrices formed by a small number of **HD** blocks can outperform non-discrete transforms.

In contrast, we are able to prove that ROMs yield improved MSE in several settings and for many of them for any number of dimensions. In addition, **SD**-product matrices can deliver computational speed benefits. We provide initial analysis to understand why $k = 3$ can outperform the state-of-the-art, why odd $k$ yields better results than even $k$, and why higher values of $k$ deliver decreasing additional benefits (see §3 and §5).

## 2 The family of Random Ortho-Matrices (ROMs)

Random ortho-matrices (ROMs) are taken from two main classes of distributions defined below that require the rows of sampled matrices to be orthogonal. A central theme of the paper is that this orthogonal structure can yield improved statistical performance. We shall use bold uppercase (e.g. **M**) to denote matrices and bold lowercase (e.g. **x**) for vectors.

**Gaussian orthogonal matrices.** Let **G** be a random matrix taking values in $\mathbb{R}^{m \times n}$ with iid $\mathcal{N}(0, 1)$ elements, which we refer to as an *unstructured* Gaussian matrix. The first ROM distribution we consider yields the random matrix $\mathbf{G}_{\text{ort}}$, which is defined as a random $\mathbb{R}^{n \times n}$ matrix given by first taking the rows of the matrix to be a uniformly random orthonormal basis, and then independently scaling each row, so that the rows marginally have multivariate Gaussian $\mathcal{N}(0, I)$ distributions. The random variable $\mathbf{G}_{\text{ort}}$ can then be extended to non-square matrices by either stacking independent copies of the $\mathbb{R}^{n \times n}$ random matrices, and deleting superfluous rows if necessary. The orthogonality of the rows of this matrix has been observed to yield improved statistical properties for randomized algorithms built from the matrix in a variety of applications.

**SD-product matrices.** Our second class of distributions is motivated by the desire to obtain similar statistical benefits of orthogonality to $\mathbf{G}_{\text{ort}}$, whilst gaining computational efficiency by employing more structured matrices. We call this second class **SD**-*product* matrices. These take the more structured form $\prod_{i=1}^{k} \mathbf{SD}_i$, where $\mathbf{S} = \{s_{i,j}\} \in \mathbb{R}^{n \times n}$ has orthogonal rows, $|s_{i,j}| = \frac{1}{\sqrt{n}} \ \forall i, j \in \{1, \ldots, n\}$; and the $(\mathbf{D}_i)_{i=1}^{k}$ are independent diagonal matrices described below. By $\prod_{i=1}^{k} \mathbf{SD}_i$, we mean the matrix product $(\mathbf{SD}_k) \ldots (\mathbf{SD}_1)$. This class includes as particular cases several recently introduced random matrices (e.g. Andoni et al., 2015; Yu et al., 2016), where good *empirical* performance was observed. We go further to establish strong theoretical guarantees, see §3 and §4.

A prominent example of an $\mathbf{S}$ matrix is the normalized *Hadamard* matrix $\mathbf{H}$, defined recursively by $\mathbf{H}_1 = (1)$, and then for $i > 1$, $\mathbf{H}_i = \frac{1}{\sqrt{2}} \begin{pmatrix} \mathbf{H}_{i-1} & \mathbf{H}_{i-1} \\ \mathbf{H}_{i-1} & -\mathbf{H}_{i-1} \end{pmatrix}$. Importantly, matrix-vector products with $\mathbf{H}$ are computable in $O(n \log n)$ time via the fast Walsh-Hadamard transform, yielding large computational savings. In addition, $\mathbf{H}$ matrices enable a significant space advantage: since the fast Walsh-Hadamard transform can be computed without explicitly storing $\mathbf{H}$, only $O(n)$ space is required to store the diagonal elements of $(\mathbf{D}_i)_{i=1}^k$. Note that these $\mathbf{H}_n$ matrices are defined only for $n$ a power of 2, but if needed, one can always adjust data by padding with 0s to enable the use of 'the next larger' $\mathbf{H}$, doubling the number of dimensions in the worst case.

Matrices $\mathbf{H}$ are representatives of a much larger family in $\mathbf{S}$ which also attains computational savings. These are $L_2$-normalized versions of Kronecker-product matrices of the form $\mathbf{A}_1 \otimes ... \otimes \mathbf{A}_l \in \mathbb{R}^{n \times n}$ for $l \in \mathbb{N}$, where $\otimes$ stands for a Kronecker product and blocks $\mathbf{A}_i \in \mathbb{R}^{d \times d}$ have entries of the same magnitude and pairwise orthogonal rows each. For these matrices, matrix-vector products are computable in $O(n(2d - 1)\log_d(n))$ time (Zhang et al., 2015).

$\mathbf{S}$ includes also the *Walsh matrices* $\mathbf{W} = \{w_{i,j}\} \in \mathbb{R}^{n \times n}$, where $w_{i,j} = \frac{1}{\sqrt{n}}(-1)^{i_{N-1}j_0 + ... + i_0 j_{N-1}}$ and $i_{N-1}...i_0$, $j_{N-1}...j_0$ are binary representations of $i$ and $j$ respectively.

For diagonal $(\mathbf{D}_i)_{i=1}^k$, we mainly consider Rademacher entries leading to the following matrices.

**Definition 2.1.** *The $\mathbf{S}$-Rademacher random matrix with $k \in \mathbb{N}$ blocks is below, where $(\mathbf{D}_i^{(\mathcal{R})})_{i=1}^k$ are diagonal with iid Rademacher random variables [i.e.* $\mathrm{Unif}(\{\pm 1\})$*] on the diagonals:*

$$\mathbf{M}_{\mathbf{S}\mathcal{R}}^{(k)} = \prod_{i=1}^k \mathbf{S}\mathbf{D}_i^{(\mathcal{R})}. \tag{1}$$

Having established the two classes of ROMs, we next apply them to dimensionality reduction.

## 3 The Orthogonal Johnson-Lindenstrauss Transform (OJLT)

Let $\mathcal{X} \subset \mathbb{R}^n$ be a dataset of $n$-dimensional real vectors. The goal of dimensionality reduction via random projections is to transform linearly each $\mathbf{x} \in \mathcal{X}$ by a random mapping $\mathbf{x} \overset{F}{\mapsto} \mathbf{x}'$, where: $F : \mathbb{R}^n \to \mathbb{R}^m$ for $m < n$, such that for any $\mathbf{x}, \mathbf{y} \in \mathcal{X}$ the following holds: $(\mathbf{x}')^\top \mathbf{y}' \approx \mathbf{x}^\top \mathbf{y}$. If we furthermore have $\mathbb{E}[(\mathbf{x}')^\top \mathbf{y}'] = \mathbf{x}^\top \mathbf{y}$ then the dot-product estimator is *unbiased*. In particular, this dimensionality reduction mechanism should in expectation preserve information about vectors' norms, i.e. we should have: $\mathbb{E}[\|\mathbf{x}'\|_2^2] = \|\mathbf{x}\|_2^2$ for any $\mathbf{x} \in \mathcal{X}$.

The standard JLT mechanism uses the randomized linear map $F = \frac{1}{\sqrt{m}}\mathbf{G}$, where $\mathbf{G} \in \mathbb{R}^{m \times n}$ is as in §2, requiring $mn$ multiplications to evaluate. Several fast variants (FJLTs) have been proposed by replacing $\mathbf{G}$ with random structured matrices, such as sparse or circulant Gaussian matrices (Ailon and Chazelle, 2006; Hinrichs and Vybíral, 2011; Vybíral, 2011; Zhang and Cheng, 2013). The fastest of these variants has $O(n \log n)$ time complexity, but at a cost of higher MSE for dot-products.

Our Orthogonal Johnson-Lindenstrauss Transform (OJLT) is obtained by replacing the unstructured random matrix $\mathbf{G}$ with a sub-sampled ROM from §2: either $\mathbf{G}_{\mathrm{ort}}$, or a sub-sampled version $\mathbf{M}_{\mathbf{S}\mathcal{R}}^{(k),\mathrm{sub}}$ of the $\mathbf{S}$-Rademacher ROM, given by sub-sampling rows from the left-most $\mathbf{S}$ matrix in the product. We sub-sample since $m < n$. We typically assume uniform sub-sampling *without* replacement. The resulting dot-product estimators for vectors $\mathbf{x}, \mathbf{y} \in \mathcal{X}$ are given by:

$$\widehat{K}_m^{\mathrm{base}}(\mathbf{x}, \mathbf{y}) = \frac{1}{m}(\mathbf{G}\mathbf{x})^\top (\mathbf{G}\mathbf{y}) \quad \text{[unstructured iid baseline, previous state-of-the-art accuracy]},$$

$$\widehat{K}_m^{\mathrm{ort}}(\mathbf{x}, \mathbf{y}) = \frac{1}{m}(\mathbf{G}_{\mathrm{ort}}\mathbf{x})^\top (\mathbf{G}_{\mathrm{ort}}\mathbf{y}), \qquad \widehat{K}_m^{(k)}(\mathbf{x}, \mathbf{y}) = \frac{1}{m}\left(\mathbf{M}_{\mathbf{S}\mathcal{R}}^{(k),\mathrm{sub}}\mathbf{x}\right)^\top \left(\mathbf{M}_{\mathbf{S}\mathcal{R}}^{(k),\mathrm{sub}}\mathbf{y}\right). \tag{2}$$

We contribute the following closed-form expressions, which exactly quantify the mean-squared error (MSE) for these three estimators. Precisely, the MSE of an estimator $\widehat{K}(\mathbf{x}, \mathbf{y})$ of the inner product $\langle \mathbf{x}, \mathbf{y} \rangle$ for $\mathbf{x}, \mathbf{y} \in \mathcal{X}$ is defined to be $\mathrm{MSE}(\widehat{K}(\mathbf{x}, \mathbf{y})) = \mathbb{E}\left[(\widehat{K}(\mathbf{x}, \mathbf{y}) - \langle \mathbf{x}, \mathbf{y} \rangle^2)\right]$. See the Appendix for detailed proofs of these results and all others in this paper.

**Lemma 3.1.** *The MSE of the unstructured JLT dot-product estimator $\widehat{K}_m^{\text{base}}$ of $\mathbf{x}, \mathbf{y} \in \mathbb{R}^n$ using $m$-dimensional random feature maps is unbiased, with* $\text{MSE}(\widehat{K}_m^{\text{base}}(\mathbf{x}, \mathbf{y})) = \frac{1}{m}((\mathbf{x}^\top \mathbf{y})^2 + \|\mathbf{x}\|_2^2 \|\mathbf{y}\|_2^2)$.

**Theorem 3.2.** *The estimator $\widehat{K}_m^{\text{ort}}$ is unbiased and satisfies, for $n \geq 4$:*

$$
\begin{aligned}
&\text{MSE}(\widehat{K}_m^{\text{ort}}(\mathbf{x}, \mathbf{y})) \\
=&\text{MSE}(\widehat{K}_m^{\text{base}}(\mathbf{x}, \mathbf{y})) + \\
&\frac{m}{m-1}\Bigg[ \frac{\|\mathbf{x}\|_2^2 \|\mathbf{y}\|_2^2 n^2}{4 I(n-3) I(n-4)} \Bigg( \left( \frac{1}{n} - \frac{1}{n+2} \right) (I(n-3) - I(n-1)) I(n-4) \left[ \cos^2(\theta) + \frac{1}{2} \right] + \\
&\qquad I(n-1) \left( I(n-4) - I(n-2) \right) \left( \frac{1}{n-2} - \frac{1}{n} \right) \left[ \cos^2(\theta) - \frac{1}{2} \right] \Bigg) - \langle \mathbf{x}, \mathbf{y} \rangle^2 \Bigg],
\end{aligned}
$$

(3)

*where $I(n) = \int_0^\pi \sin^n(x)dx = \frac{\sqrt{\pi}\Gamma((n+1)/2)}{\Gamma(n/2+1)}$.*

**Theorem 3.3** (Key result). *The OJLT estimator $\widehat{K}_m^{(k)}(\mathbf{x}, \mathbf{y})$ with $k$ blocks, using $m$-dimensional random feature maps and uniform sub-sampling policy without replacement, is unbiased with*

$$
\text{MSE}(\widehat{K}_m^{(k)}(\mathbf{x}, \mathbf{y})) = \frac{1}{m} \left( \frac{n-m}{n-1} \right) \Bigg( ((\mathbf{x}^\top \mathbf{y})^2 + \|\mathbf{x}\|^2 \|\mathbf{y}\|^2) + \tag{4}
$$
$$
\sum_{r=1}^{k-1} \frac{(-1)^r 2^r}{n^r} (2(\mathbf{x}^\top \mathbf{y})^2 + \|\mathbf{x}\|^2 \|\mathbf{y}\|^2) + \frac{(-1)^k 2^k}{n^{k-1}} \sum_{i=1}^{n} x_i^2 y_i^2 \Bigg).
$$

*Proof (Sketch).* For $k = 1$, the random projection matrix is given by sub-sampling rows from $\mathbf{SD}_1$, and the computation can be carried out directly. For $k \geq 1$, the proof proceeds by induction. The random projection matrix in the general case is given by sub-sampling rows of the matrix $\mathbf{SD}_k \cdots \mathbf{SD}_1$. By writing the MSE as an expectation and using the law of conditional expectations conditioning on the value of the first $k-1$ random matrices $\mathbf{D}_{k-1}, \ldots, \mathbf{D}_1$, the statement of the theorem for 1 $\mathbf{SD}$ block and for $k-1$ $\mathbf{SD}$ blocks can be neatly combined to yield the result. $\square$

To our knowledge, it has not previously been possible to provide theoretical guarantees that $\mathbf{SD}$-product matrices outperform iid matrices. Combining Lemma 3.1 with Theorem 3.3 yields the following important result.

**Corollary 3.4** (Theoretical guarantee of improved performance). *Estimators $\widehat{K}_m^{(k)}$ (subsampling without replacement) yield guaranteed lower MSE than $\widehat{K}_m^{\text{base}}$.*

It is not yet clear when $\widehat{K}_m^{\text{ort}}$ is better or worse than $\widehat{K}_m^{(k)}$; we explore this empirically in §6. Theorem 3.3 shows that there are diminishing MSE benefits to using a large number $k$ of $\mathbf{SD}$ blocks. Interestingly, odd $k$ is better than even: it is easy to observe that $\text{MSE}(\widehat{K}_m^{(2k-1)}(\mathbf{x}, \mathbf{y})) < \text{MSE}(\widehat{K}_m^{(2k)}(\mathbf{x}, \mathbf{y})) > \text{MSE}(\widehat{K}_m^{(2k+1)}(\mathbf{x}, \mathbf{y}))$. These observations, and those in §5, help to understand why empirically $k = 3$ was previously observed to work well (Yu et al., 2016).

If we take $\mathbf{S}$ to be a normalized Hadamard matrix $\mathbf{H}$, then even though we are using sub-sampling, and hence the full computational benefits of the Walsh-Hadamard transform are not available, still $\widehat{K}_m^{(k)}$ achieves improved MSE compared to the base method with *less* computational effort, as follows.

**Lemma 3.5.** *There exists an algorithm (see Appendix for details) which computes an embedding for a given datapoint $\mathbf{x}$ using $\widehat{K}_m^{(k)}$ with $\mathbf{S}$ set to $\mathbf{H}$ and uniform sub-sampling policy in expected time* $\min\{O((k-1)n \log(n) + nm - \frac{(m-1)m}{2}, kn \log(n)\}$.

Note that for $m = \omega(k \log(n))$ or if $k = 1$, the time complexity is smaller than the brute force $\Theta(nm)$. The algorithm uses a simple observation that one can reuse calculations conducted for the upper half of the Hadamard matrix while performing computations involving rows from its other half, instead of running these calculations from scratch (details in the Appendix).

An alternative to sampling without replacement is deterministically to choose the first $m$ rows. In our experiments in §6, these two approaches yield the same empirical performance, though we expect

that the deterministic method could perform poorly on adversarially chosen data. The first $m$ rows approach can be realized in time $O(n \log(m) + (k-1)n \log(n))$ per datapoint.

Theorem 3.3 is a key result in this paper, demonstrating that **SD**-product matrices yield both statistical and computational improvements compared to the base iid procedure, which is widely used in practice. We next show how to obtain further gains in accuracy.

### 3.1 Complex variants of the OJLT

We show that the MSE benefits of Theorem 3.3 may be markedly improved by using **SD**-product matrices with complex entries $\mathbf{M}_{\mathbf{S}\mathcal{H}}^{(k)}$. Specifically, we consider the variant **S**-*Hybrid* random matrix below, where $\mathbf{D}_k^{(\mathcal{U})}$ is a diagonal matrix with iid $\mathrm{Unif}(S^1)$ random variables on the diagonal, independent of $(\mathbf{D}_i^{(\mathcal{R})})_{i=1}^{k-1}$, and $S^1$ is the unit circle of $\mathbb{C}$. We use the real part of the Hermitian product between projections as a dot-product estimator; recalling the definitions of §2, we use:

$$\mathbf{M}_{\mathbf{S}\mathcal{H}}^{(k)} = \mathbf{SD}_k^{(\mathcal{U})} \prod_{i=1}^{k-1} \mathbf{SD}_i^{(\mathcal{R})}, \qquad \widehat{K}_m^{\mathcal{H},(k)}(\mathbf{x}, \mathbf{y}) = \frac{1}{m} \mathrm{Re} \left[ \left( \overline{\mathbf{M}_{\mathbf{S}\mathcal{H}}^{(k),\mathrm{sub}} \mathbf{x}} \right)^\top \left( \mathbf{M}_{\mathbf{S}\mathcal{H}}^{(k),\mathrm{sub}} \mathbf{y} \right) \right]. \quad (5)$$

Remarkably, this complex variant yields exactly half the MSE of the OJLT estimator.

**Theorem 3.6.** *The estimator $\widehat{K}_m^{\mathcal{H},(k)}(\mathbf{x}, \mathbf{y})$, applying uniform sub-sampling without replacement, is unbiased and satisfies:* $\mathrm{MSE}(\widehat{K}_m^{\mathcal{H},(k)}(\mathbf{x}, \mathbf{y})) = \frac{1}{2}\mathrm{MSE}(\widehat{K}_m^{(k)}(\mathbf{x}, \mathbf{y}))$.

This large factor of 2 improvement could instead be obtained by doubling $m$ for $\widehat{K}_m^{(k)}$. However, this would require doubling the number of parameters for the transform, whereas the **S**-Hybrid estimator requires additional storage only for the complex parameters in the matrix $\mathbf{D}_k^{(\mathcal{U})}$. Strikingly, it is straightforward to extend the proof of Theorem 3.6 (see Appendix) to show that rather than taking the complex random variables in $\mathbf{M}_{\mathbf{S}\mathcal{H}}^{(k),\mathrm{sub}}$ to be $\mathrm{Unif}(S^1)$, it is possible to take them to be $\mathrm{Unif}(\{1, -1, i, -i\})$ and still obtain exactly the same benefit in MSE.

**Theorem 3.7.** *For the estimator $\widehat{K}_m^{\mathcal{H},(k)}$ defined in Equation (5): replacing the random matrix $\mathbf{D}_k^{(\mathcal{U})}$ (which has iid $\mathrm{Unif}(S^1)$ elements on the diagonal) with instead a random diagonal matrix having iid $\mathrm{Unif}(\{1, -1, i, -i\})$ elements on the diagonal, does not affect the MSE of the estimator.*

It is natural to wonder if using an **SD**-product matrix with more complex random variables (for all **SD** blocks) would improve performance still further. However, interestingly, this appears not to be the case; details are provided in the Appendix §8.7.

### 3.2 Sub-sampling with replacement

Our results above focus on **SD**-product matrices where rows have been sub-sampled without replacement. Sometimes (e.g. for parallelization) it can be convenient instead to sub-sample *with* replacement. As might be expected, this leads to worse MSE, which we can quantify precisely.

**Theorem 3.8.** *For each of the estimators $\widehat{K}_m^{(k)}$ and $\widehat{K}_m^{\mathcal{H},(k)}$, if uniform sub-sampling* with *(rather than without) replacement is used then the MSE is worsened by a multiplicative constant of $\frac{n-1}{n-m}$.*

## 4 Kernel methods with ROMs

ROMs can also be used to construct high-quality random feature maps for non-linear kernel approximation. We analyze here the *angular kernel*, an important example of a *Pointwise Nonlinear Gaussian kernel* (PNG), discussed in more detail at the end of this section.

**Definition 4.1.** *The angular kernel $K^{\mathrm{ang}}$ is defined on $\mathbb{R}^n$ by $K^{\mathrm{ang}}(\mathbf{x}, \mathbf{y}) = 1 - \frac{2\theta_{\mathbf{x},\mathbf{y}}}{\pi}$, where $\theta_{\mathbf{x},\mathbf{y}}$ is the angle between $\mathbf{x}$ and $\mathbf{y}$.*

To employ random feature style approximations to this kernel, we first observe it may be rewritten as

$$K^{\mathrm{ang}}(\mathbf{x}, \mathbf{y}) = \mathbb{E}\left[\mathrm{sign}(\mathbf{Gx})\mathrm{sign}(\mathbf{Gy})\right],$$

where $\mathbf{G} \in \mathbb{R}^{1 \times n}$ is an unstructured isotropic Gaussian vector. This motivates approximations of the form:

$$\widehat{K}^{\mathrm{ang}}m(\mathbf{x}, \mathbf{y}) = \frac{1}{m}\mathrm{sign}(\mathbf{Mx})^{\top}\mathrm{sign}(\mathbf{My}), \tag{6}$$

where $\mathbf{M} \in \mathbb{R}^{m \times n}$ is a random matrix, and the sign function is applied coordinate-wise. Such kernel estimation procedures are heavily used in practice (Rahimi and Recht, 2007), as they allow fast approximate linear methods to be used (Joachims, 2006) for inference tasks. If $\mathbf{M} = \mathbf{G}$, the unstructured Gaussian matrix, then we obtain the standard random feature estimator. We shall contrast this approach against the use of matrices from the ROMs family.

When constructing random feature maps for kernels, very often $m > n$. In this case, our structured mechanism can be applied by concatenating some number of independent structured blocks. Our theoretical guarantees will be given just for one block, but can easily be extended to a larger number of blocks since different blocks are independent.

The standard random feature approximation $\widehat{K}_m^{\mathrm{ang,base}}$ for approximating the angular kernel is defined by taking $\mathbf{M}$ to be $\mathbf{G}$, the unstructured Gaussian matrix, in Equation (6), and satisfies the following.

**Lemma 4.2.** *The estimator* $\widehat{K}_m^{\mathrm{ang,base}}$ *is unbiased and* $\mathrm{MSE}(\widehat{K}_m^{\mathrm{ang,base}}(\mathbf{x}, \mathbf{y})) = \frac{4\theta_{\mathbf{x},\mathbf{y}}(\pi - \theta_{\mathbf{x},\mathbf{y}})}{m\pi^2}$.

The MSE of an estimator $\widehat{K}^{\mathrm{ang}}(\mathbf{x}, \mathbf{y})$ of the true angular kernel $K^{\mathrm{ang}}(\mathbf{x}, \mathbf{y})$ is defined analogously to the MSE of an estimator of the dot product, given in §3. Our main result regarding angular kernels states that if we instead take $\mathbf{M} = \mathbf{G}_{\mathrm{ort}}$ in Equation (6), then we obtain an estimator $\widetilde{K}_m^{\mathrm{ang,ort}}$ with strictly smaller MSE, as follows.

**Theorem 4.3.** *Estimator* $\widehat{K}_m^{\mathrm{ang,ort}}$ *is unbiased and satisfies:*

$$\mathrm{MSE}(\widehat{K}_m^{\mathrm{ang,ort}}(\mathbf{x}, \mathbf{y})) < \mathrm{MSE}(\widehat{K}_m^{\mathrm{ang,base}}(\mathbf{x}, \mathbf{y})).$$

We also derive a formula for the MSE of an estimator $\widehat{K}_m^{\mathrm{ang,M}}$ of the angular kernel which replaces $\mathbf{G}$ with an arbitrary random matrix $\mathbf{M}$ and uses $m$ random feature maps. The formula is helpful to see how the quality of the estimator depends on the probabilities that the projections of the rows of $\mathbf{M}$ are contained in some particular convex regions of the 2-dimensional space $\mathcal{L}_{\mathbf{x},\mathbf{y}}$ spanned by datapoints $\mathbf{x}$ and $\mathbf{y}$. For an illustration of the geometric definitions introduced in this Section, see Figure 1. The formula depends on probabilities involving events $\mathcal{A}^i = \{\mathrm{sgn}((\mathbf{r}^i)^T\mathbf{x}) \neq \mathrm{sgn}((\mathbf{r}^i)^T\mathbf{y})\}$, where $\mathbf{r}^i$ stands for the $i^{th}$ row of the structured matrix. Notice that $\mathcal{A}^i = \{\mathbf{r}_{proj}^i \in \mathcal{C}_{\mathbf{x},\mathbf{y}}\}$, where $\mathbf{r}_{proj}^i$ stands for the projection of $\mathbf{r}^i$ into $\mathcal{L}_{\mathbf{x},\mathbf{y}}$ and $\mathcal{C}_{\mathbf{x},\mathbf{y}}$ is the union of two cones in $\mathcal{L}_{\mathbf{x},\mathbf{y}}$, each of angle $\theta_{\mathbf{x},\mathbf{y}}$.

**Theorem 4.4.** *Estimator* $\widehat{K}_m^{\mathrm{ang,M}}$ *satisfies the following, where:* $\delta_{i,j} = \mathbb{P}[\mathcal{A}^i \cap \mathcal{A}^j] - \mathbb{P}[\mathcal{A}^i]\mathbb{P}[\mathcal{A}^j]$:

$$\mathrm{MSE}(\widehat{K}_m^{\mathrm{ang,M}}(\mathbf{x}, \mathbf{y})) = \frac{1}{m^2}\left[m - \sum_{i=1}^m (1 - 2\mathbb{P}[\mathcal{A}^i])^2\right] + \frac{4}{m^2}\left[\sum_{i=1}^m (\mathbb{P}[\mathcal{A}^i] - \frac{\theta_{\mathbf{x},\mathbf{y}}}{\pi})^2 + \sum_{i \neq j}\delta_{i,j}\right].$$

Note that probabilities $\mathbb{P}[\mathcal{A}^i]$ and $\delta_{i,j}$ depend on the choice of $\mathbf{M}$. It is easy to prove that for unstructured $\mathbf{G}$ and $\mathbf{G}_{\mathrm{ort}}$ we have: $\mathbb{P}[\mathcal{A}^i] = \frac{\theta_{\mathbf{x},\mathbf{y}}}{\pi}$. Further, from the independence of the rows of $\mathbf{G}$, $\delta_{i,j} = 0$ for $i \neq j$. For unstructured $\mathbf{G}$ we obtain Lemma 4.2. Interestingly, we see that to prove Theorem 4.3, it suffices to show $\delta_{i,j} < 0$, which is the approach we take (see Appendix). If we replace $\mathbf{G}$ with $\mathbf{M}_{S\mathcal{R}}^{(k)}$, then the expression $\epsilon = \mathbb{P}[\mathcal{A}^i] - \frac{\theta_{\mathbf{x},\mathbf{y}}}{\pi}$ does not depend on $i$. Hence, the angular kernel estimator based on Hadamard matrices gives smaller MSE estimator if and only if $\sum_{i \neq j}\delta_{i,j} + m\epsilon^2 < 0$. It is not yet clear if this holds in general.

As alluded to at the beginning of this section, the angular kernel may be viewed as a member of a wie family of kernels known as Pointwise Nonlinear Gaussian kernels.

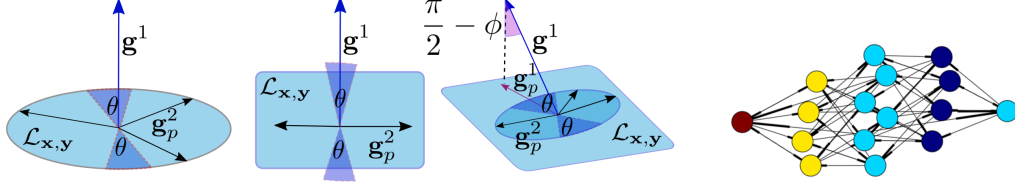

Figure 1: **Left part:** Left: $\mathbf{g}^1$ is orthogonal to $\mathcal{L}_{\mathbf{x},\mathbf{y}}$. Middle: $\mathbf{g}^1 \in \mathcal{L}_{\mathbf{x},\mathbf{y}}$. Right: $\mathbf{g}^1$ is close to orthogonal to $\mathcal{L}_{\mathbf{x},\mathbf{y}}$. **Right part:** Visualization of the Cayley graph explored by the Hadamard-Rademacher process in two dimensions. Nodes are colored red, yellow, light blue, dark blue, for Cayley distances of $0, 1, 2, 3$ from the identity matrix respectively. See text in §5.

**Definition 4.5.** *For a given function $f$, the Pointwise Nonlinear Gaussian kernel (PNG) $K^f$ is defined by $K^f(\mathbf{x}, \mathbf{y}) = \mathbb{E}\left[f(\mathbf{g}^T\mathbf{x})f(\mathbf{g}^T\mathbf{y})\right]$, where $\mathbf{g}$ is a Gaussian vector with i.i.d $\mathcal{N}(0,1)$ entries.*

Many prominent examples of kernels (Williams, 1998; Cho and Saul, 2009) are PNGs. Wiener's tauberian theorem shows that all stationary kernels may be approximated arbitrarily well by sums of PNGs (Samo and Roberts, 2015). In future work we hope to explore whether ROMs can be used to achieve statistical benefit in estimation tasks associated with a wider range of PNGs.

# 5 Understanding the effectiveness of orthogonality

Here we build intuitive understanding for the effectiveness of ROMs. We examine geometrically the angular kernel (see §4), then discuss a connection to random walks over orthogonal matrices.

**Angular kernel.** As noted above for the $\mathbf{G}_{\text{ort}}$-mechanism, smaller MSE than that for unstructured $\mathbf{G}$ is implied by the inequality $\mathbb{P}[\mathcal{A}^i \cap \mathcal{A}^j] < \mathbb{P}[\mathcal{A}^i]\mathbb{P}[\mathcal{A}^j]$, which is equivalent to: $\mathbb{P}[\mathcal{A}^j|\mathcal{A}^i] < \mathbb{P}[\mathcal{A}^j]$. Now it becomes clear why orthogonality is crucial. Without loss of generality take: $i = 1$, $j = 2$, and let $\mathbf{g}^1$ and $\mathbf{g}^2$ be the first two rows of $\mathbf{G}_{\text{ort}}$.

Consider first the extreme case (middle of left part of Figure 1), where all vectors are 2-dimensional. Recall definitions from just after Theorem 4.3. If $\mathbf{g}^1$ is in $\mathcal{C}_{\mathbf{x},\mathbf{y}}$ then it is much less probable for $\mathbf{g}^2$ also to belong to $\mathcal{C}_{\mathbf{x},\mathbf{y}}$. In particular, if $\theta < \frac{\pi}{2}$ then the probability is zero. That implies the inequality. On the other hand, if $\mathbf{g}^1$ is perpendicular to $\mathcal{L}_{\mathbf{x},\mathbf{y}}$ then conditioning on $\mathcal{A}^i$ does not have any effect on the probability that $\mathbf{g}^2$ belongs to $\mathcal{C}_{\mathbf{x},\mathbf{y}}$ (left subfigure of Figure 1). In practice, with high probability the angle $\phi$ between $\mathbf{g}^1$ and $\mathcal{L}_{\mathbf{x},\mathbf{y}}$ is close to $\frac{\pi}{2}$, but is not exactly $\frac{\pi}{2}$. That again implies that conditioned on the projection $\mathbf{g}_p^1$ of $\mathbf{g}^1$ into $\mathcal{L}_{\mathbf{x},\mathbf{y}}$ to be in $\mathcal{C}_{\mathbf{x},\mathbf{y}}$, the more probable directions of $\mathbf{g}_p^2$ are perpendicular to $\mathbf{g}_p^1$ (see: ellipsoid-like shape in the right subfigure of Figure 1 which is the projection of the sphere taken from the $(n-1)$-dimensional space orthogonal to $\mathbf{g}^1$ into $\mathcal{L}_{\mathbf{x},\mathbf{y}}$). This makes it less probable for $\mathbf{g}_p^2$ to be also in $\mathcal{C}_{\mathbf{x},\mathbf{y}}$. The effect is subtle since $\phi \approx \frac{\pi}{2}$, but this is what provides superiority of the orthogonal transformations over state-of-the-art ones in the angular kernel approximation setting.

**Markov chain perspective.** We focus on Hadamard-Rademacher random matrices $\mathbf{HD}_k...\mathbf{HD}_1$, a special case of the $\mathbf{SD}$-product matrices described in Section 2. Our aim is to provide intuition for how the choice of $k$ affects the quality of the random matrix, following our earlier observations just after Corollary 3.4, which indicated that for $\mathbf{SD}$-product matrices, odd values of $k$ yield greater benefits than even values, and that there are diminishing benefits from higher values of $k$. We proceed by casting the random matrices into the framework of Markov chains.

**Definition 5.1.** *The Hadamard-Rademacher process in $n$ dimensions is the Markov chain $(\mathbf{X}_k)_{k=0}^\infty$ taking values in the orthogonal group $O(n)$, with $\mathbf{X}_0 = \mathbf{I}$ almost surely, and $\mathbf{X}_k = \mathbf{HD}_k\mathbf{X}_{k-1}$ almost surely, where $\mathbf{H}$ is the normalized Hadamard matrix in $n$ dimensions, and $(\mathbf{D}_k)_{k=1}^\infty$ are iid diagonal matrices with independent Rademacher random variables on their diagonals.*

Constructing an estimator based on Hadamard-Rademacher matrices is equivalent to simulating several time steps from the Hadamard-Rademacher process. The quality of estimators based on Hadamard-Rademacher random matrices comes from a quick mixing property of the corresponding

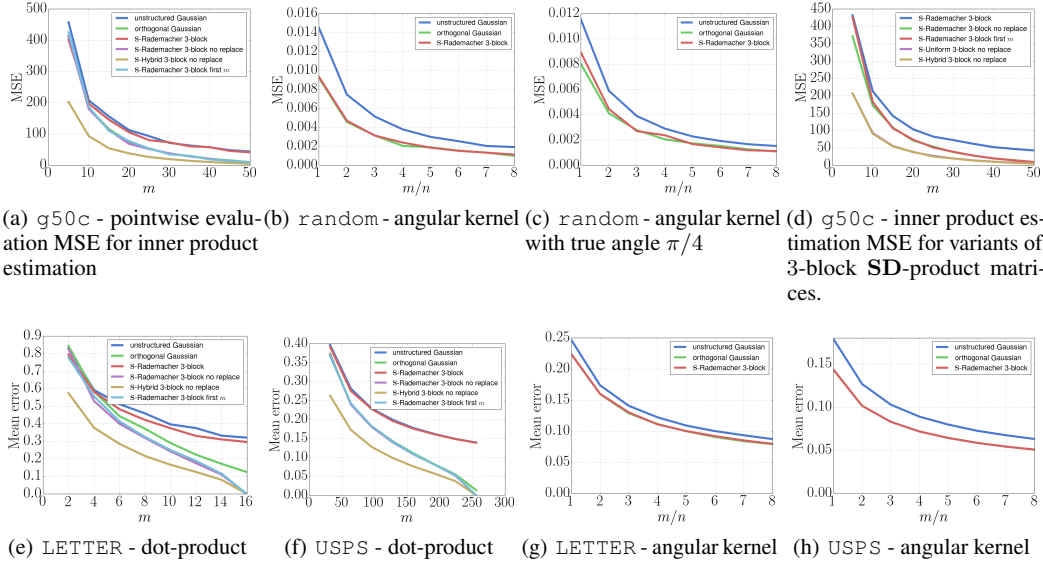

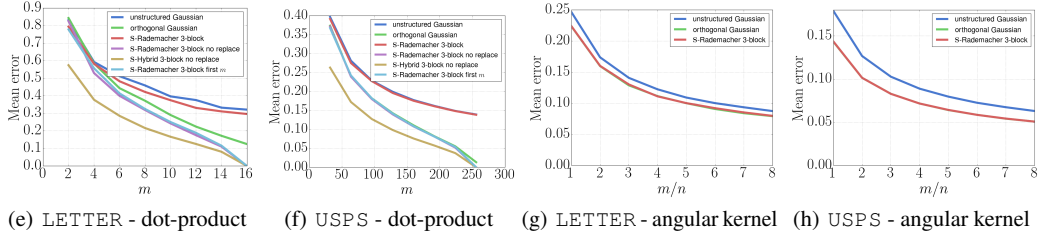

(a) `g50c` - pointwise evalu- (b) `random` - angular kernel (c) `random` - angular kernel (d) `g50c` - inner product es-
ation MSE for inner product with true angle $\pi/4$ timation MSE for variants of
estimation 3-block **SD**-product matri-
ces.

(e) `LETTER` - dot-product (f) `USPS` - dot-product (g) `LETTER` - angular kernel (h) `USPS` - angular kernel

Figure 2: **Top row:** MSE curves for pointwise approximation of inner product and angular kernels on the `g50c` dataset, and randomly chosen vectors. **Bottom row:** Gram matrix approximation error for a variety of data sets, projection ranks, transforms, and kernels. Note that the error scaling is dependent on the application.

Markov chain. The following demonstrates attractive properties of the chain in low dimensions.

**Proposition 5.2.** *The Hadamard-Rademacher process in two dimensions: explores a state-space of* 16 *orthogonal matrices, is ergodic with respect to the uniform distribution on this set, has period* 2, *the diameter of the Cayley graph of its state space is* 3, *and the chain is fully mixed after* 3 *time steps.*

This proposition, and the Cayley graph corresponding to the Markov chain's state space (Figure 1 right), illustrate the fast mixing properties of the Hadamard-Rademacher process in low dimensions; this agrees with the observations in §3 that there are diminishing returns associated with using a large number $k$ of **HD** blocks in an estimator. The observation in Proposition 5.2 that the Markov chain has period 2 indicates that we should expect different behavior for estimators based on odd and even numbers of blocks of **HD** matrices, which is reflected in the analytic expressions for MSE derived in Theorems 3.3 and 3.6 for the dimensionality reduction setup.

## 6 Experiments

We present comparisons of estimators introduced in §3 and §4, illustrating our theoretical results, and further demonstrating the empirical success of ROM-based estimators at the level of Gram matrix approximation. We compare estimators based on: unstructured Gaussian matrices **G**, matrices **G**$_{\text{ort}}$, **S**-*Rademacher* and **S**-*Hybrid* matrices with $k = 3$ and different sub-sampling strategies. Results for $k > 3$ do not show additional statistical gains empirically. Additional experimental results, including a comparison of estimators using different numbers of **SD** blocks, are in the Appendix §10. Throughout, we use the normalized Hadamard matrix **H** for the structured matrix **S**.

### 6.1 Pointwise kernel approximation

Complementing the theoretical results of §3 and §4, we provide several salient comparisons of the various methods introduced - see Figure 2 top. Plots presented here (and in the Appendix) compare MSE for dot-product and angular and kernel. They show that estimators based on **G**$_{\text{ort}}$, **S**-*Hybrid* and **S**-*Rademacher* matrices without replacement, or using the first $m$ rows, beat the state-of-the-art unstructured **G** approach on accuracy for all our different datasets in the JLT setup. Interestingly, the latter two approaches give also smaller MSE than **G**$_{\text{ort}}$-estimators. For angular kernel estimation, where sampling is not relevant, we see that **G**$_{\text{ort}}$ and **S**-*Rademacher* approaches again outperform the ones based on matrices **G**.

## 6.2 Gram matrix approximation

Moving beyond the theoretical guarantees established in §3 and §4, we show empirically that the superiority of estimators based on ROMs is maintained at the level of Gram matrix approximation. We compute Gram matrix approximations (with respect to both standard dot-product, and angular kernel) for a variety of datasets. We use the normalized Frobenius norm error $\|\mathbf{K} - \widehat{\mathbf{K}}\|_2 / \|\mathbf{K}\|_2$ as our metric (as used by Choromanski and Sindhwani, 2016), and plot the mean error based on 1,000 repetitions of each random transform - see Figure 2 bottom. The Gram matrices are computed on a randomly selected subset of 550 data points from each dataset. As can be seen, the $\mathbf{S}$-*Hybrid* estimators using the "no-replacement" or "first $m$ rows" sub-sampling strategies outperform even the orthogonal Gaussian ones in the dot-product case. For the angular case, the $\mathbf{G}_{\text{ort}}$-approach and $\mathbf{S}$-*Rademacher* approach are practically indistinguishable.

## 7 Conclusion

We defined the family of random ortho-matrices (ROMs). This contains the $\mathbf{SD}$-product matrices, which include a number of recently proposed structured random matrices. We showed theoretically and empirically that ROMs have strong statistical and computational properties (in several cases outperforming previous state-of-the-art) for algorithms performing dimensionality reduction and random feature approximations of kernels. We highlight Corollary 3.4, which provides a theoretical guarantee that $\mathbf{SD}$-product matrices yield better accuracy than iid matrices in an important dimensionality reduction application (we believe the first result of this kind). Intriguingly, for dimensionality reduction, using just one complex structured matrix yields random features of much better quality. We provided perspectives to help understand the benefits of ROMs, and to help explain the behavior of $\mathbf{SD}$-product matrices for various numbers of blocks. Our empirical findings suggest that our theoretical results might be further strengthened, particularly in the kernel setting.

## Acknowledgements

We thank Vikas Sindhwani at Google Brain Robotics and Tamas Sarlos at Google Research for inspiring conversations that led to this work. We thank Matej Balog, Maria Lomeli, Jiri Hron and Dave Janz for helpful comments. MR acknowledges support by the UK Engineering and Physical Sciences Research Council (EPSRC) grant EP/L016516/1 for the University of Cambridge Centre for Doctoral Training, the Cambridge Centre for Analysis. AW acknowledges support by the Alan Turing Institute under the EPSRC grant EP/N510129/1, and by the Leverhulme Trust via the CFI.

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
