[Supplementary Material]

# APPENDIX:
# The Unreasonable Effectiveness of Random Orthogonal Embeddings

We present here details and proofs of all the theoretical results presented in the main body of the paper. We also provide further experimental results in §10.

We highlight proofs of several key results that may be of particular interest to the reader:

- The proof of Theorem 3.3; see §8.3.
- The proof of Theorem 3.6; see §8.5.
- The proof of Theorem 4.3; see §9.2.

In the Appendix we will use interchangeably two notations for the dot product between vectors $\mathbf{x}$ and $\mathbf{y}$, namely: $\mathbf{x}^\top \mathbf{y}$ and $\langle \mathbf{x}, \mathbf{y} \rangle$.

## 8 Proofs of results in §3

### 8.1 Proof of Lemma 3.1

*Proof.* Denote $X_i = (\mathbf{g}^i)^\top \mathbf{x} \cdot (\mathbf{g}^i)^\top \mathbf{y}$, where $\mathbf{g}^i$ stands for the $i^{th}$ row of the unstructured Gaussian matrix $\mathbf{G} \in \mathbb{R}^{m \times n}$. Note that we have:

$$\widehat{K}_m^{\text{base}}(\mathbf{x}, \mathbf{y}) = \frac{1}{m} \sum_{i=1}^m X_i. \tag{7}$$

Denote $\mathbf{g}^i = (g_1^i, ..., g_n^i)^\top$. Notice that from the independence of $g_j^i$s and the fact that: $\mathbb{E}[g_j^i] = 0$, $\mathbb{E}[(g_j^i)^2] = 1$, we get: $\mathbb{E}[X_i] = \sum_{i=1}^n x_i y_i = \mathbf{x}^\top \mathbf{y}$, thus the estimator is unbiased. Since the estimator is unbiased, we have: $\text{MSE}(\widehat{K}_m^{\text{base}}(\mathbf{x}, \mathbf{y})) = Var(\widehat{K}_m^{\text{base}}(\mathbf{x}, \mathbf{y}))$. Thus we get:

$$\text{MSE}(\widehat{K}_m^{\text{base}}(\mathbf{x}, \mathbf{y})) = \frac{1}{m^2} \sum_{i,j} (\mathbb{E}[X_i X_j] - \mathbb{E}[X_i]\mathbb{E}[X_j]). \tag{8}$$

From the independence of different $X_i$s, we get:

$$\text{MSE}(\widehat{K}_m^{\text{base}}(\mathbf{x}, \mathbf{y})) = \frac{1}{m^2} \sum_i (\mathbb{E}[X_i^2] - (\mathbb{E}[X_i])^2). \tag{9}$$

Now notice that different $X_i$s have the same distribution, thus we get:

$$\text{MSE}(\widehat{K}_m^{\text{base}}(\mathbf{x}, \mathbf{y})) = \frac{1}{m} (\mathbb{E}[X_1^2] - (\mathbb{E}[X_1])^2). \tag{10}$$

From the unbiasedness of the estimator, we have: $\mathbb{E}[X_1] = \mathbf{x}^\top \mathbf{y}$. Therefore we obtain:

$$\text{MSE}(\widehat{K}_m^{\text{base}}(\mathbf{x}, \mathbf{y})) = \frac{1}{m} (\mathbb{E}[X_1^2] - (\mathbf{x}^\top \mathbf{y})^2). \tag{11}$$

Now notice that

$$\mathbb{E}[X_1^2] = \mathbb{E}[\sum_{i_1, j_1, i_2, j_2} g_{i_1} g_{j_1} g_{i_2} g_{j_2} x_{i_1} y_{j_1} x_{i_2} y_{j_2}] = \sum_{i_1, j_1, i_2, j_2} x_{i_1} y_{j_1} x_{i_2} y_{j_2} \mathbb{E}[g_{i_1} g_{j_1} g_{i_2} g_{j_2}], \tag{12}$$

where $(g_1, ..., g_n)$ stands for the first row of $\mathbf{G}$. In the expression above the only nonzero terms corresponds to quadruples $(i_1, j_1, i_2, j_2)$, where no index appears odd number of times. Therefore, from the inclusion-exclusion principle and the fact that $\mathbb{E}[g_i^2] = 1$ and $\mathbb{E}[g_i^4] = 3$, we obtain

$$\mathbb{E}[X_1^2] = \sum_{i_1 = j_1, i_2 = j_2} x_{i_1} y_{j_1} x_{i_2} y_{j_2} \mathbb{E}[g_{i_1} g_{j_1} g_{i_2} g_{j_2}] + \sum_{i_1 = i_2, j_1 = j_2} x_{i_1} y_{j_1} x_{i_2} y_{j_2} \mathbb{E}[g_{i_1} g_{j_1} g_{i_2} g_{j_2}] \tag{13}$$

$$+ \sum_{i_1=j_2, i_2=j_1} x_{i_1} y_{j_1} x_{i_2} y_{j_2} \mathbb{E}[g_{i_1} g_{j_1} g_{i_2} g_{j_2}] - \sum_{i_1=j_1=i_2=j_2} x_{i_1} y_{j_1} x_{i_2} y_{j_2} \mathbb{E}[g_{i_1} g_{j_1} g_{i_2} g_{j_2}] \tag{14}$$

$$= ((\mathbf{x}^\top \mathbf{y})^2 - \sum_{i=1}^n x_i^2 y_i^2 + 3 \sum_{i=1}^n x_i^2 y_i^2) + ((\|\mathbf{x}\|_2 \|\mathbf{y}\|_2)^2 - \sum_{i=1}^n x_i^2 y_i^2 + 3 \sum_{i=1}^n x_i^2 y_i^2) \tag{15}$$

$$+ ((\mathbf{x}^\top \mathbf{y})^2 - \sum_{i=1}^n x_i^2 y_i^2 + 3 \sum_{i=1}^n x_i^2 y_i^2) - 3 \cdot 2 \sum_{i=1}^n x_i^2 y_i^2 \tag{16}$$

$$= (\|\mathbf{x}\|_2 \|\mathbf{y}\|_2)^2 + 2(\mathbf{x}^\top \mathbf{y})^2. \tag{17}$$

Therefore we obtain

$$\mathrm{MSE}(\widehat{K}_m^{\text{base}}(\mathbf{x}, \mathbf{y})) = \frac{1}{m}((\|\mathbf{x}\|_2 \|\mathbf{y}\|_2)^2 + 2(\mathbf{x}^\top \mathbf{y})^2 - (\mathbf{x}^\top \mathbf{y})^2) = \frac{1}{m}(\|\mathbf{x}\|_2^2 \|\mathbf{y}\|_2^2 + (\mathbf{x}^\top \mathbf{y})^2), \tag{18}$$

which completes the proof. $\qquad\square$

## 8.2 Proof of Theorem 3.2

*Proof.* The unbiasedness of the Gaussian orthogonal estimator comes from the fact that every row of the Gaussian orthogonal matrix is sampled from multivariate Gaussian distribution with entries taken independently at random from $\mathcal{N}(0, 1)$.

Note that:
$$\mathrm{Cov}(X_i, X_j) = \mathbb{E}[X_i X_j] - \mathbb{E}[X_i]\mathbb{E}[X_j], \tag{19}$$
where: $X_i = (\mathbf{r}_i^\top \mathbf{x})(\mathbf{r}_i^\top \mathbf{y})$, $X_j = (\mathbf{r}_j^\top \mathbf{x})(\mathbf{r}_j^\top \mathbf{y})$ and $\mathbf{r}_i, \mathbf{r}_j$ stand for the $i^{th}$ and $j^{th}$ row of the Gaussian orthogonal matrix respectively. From the fact that Gaussian orthogonal estimator is unbiased, we get:
$$\mathbb{E}[X_i] = \mathbf{x}^\top \mathbf{y}. \tag{20}$$

Let us now compute $\mathbb{E}[X_i X_j]$. Writing $\mathbf{Z}_1 = \mathbf{r}_i$, $\mathbf{Z}_2 = \mathbf{r}_j$, we begin with some geometric observations:

- If $\phi \in [0, \pi/2]$ is the acute angle between $\mathbf{Z}_1$ and the **x-y** plane, then $\phi$ has density $f(\phi) = (n-2)\cos(\phi)\sin^{n-3}(\phi)$.

- The squared norm of the projection of $\mathbf{Z}_1$ into the **x-y** plane is therefore given by the product of a $\chi_n^2$ random variable (the norm of $\mathbf{Z}_2$), multiplied by $\cos^2(\phi)$, where $\phi$ is distributed as described above, independently from the $\chi_n^2$ random variable.

- The angle $\psi \in [0, 2\pi)$ between **x** and the projection of $\mathbf{Z}_1$ into the **x-y** plane is distributed uniformly.

- Conditioned on the angle $\phi$, the direction of $\mathbf{Z}_2$ is distributed uniformly on the hyperplane of $\mathbb{R}^n$ orthogonal to $\mathbf{Z}_1$. Using hyperspherical coordinates for the unit hypersphere of this hyperplane, we may pick an orthonormal basis of the **x-y** plane such that the first basis vector is the unit vector in the direction of the projection of $\mathbf{Z}_1$, and the coordinates of the projection of $\mathbf{Z}_2$ with respect to this basis are $(\sin(\phi)\cos(\varphi_1), \sin(\varphi_1)\cos(\varphi_2))$, where $\varphi_1, \varphi_2$ are random angles taking values in $[0, \pi]$, with densities given by $\sin^{n-3}(\varphi_1)I(n-3)^{-1}$ and $\sin^{n-4}(\varphi_2)I(n-4)^{-1}$ respectively. Here $I(k) = \int_0^\pi \sin^k(x)dx = \sqrt{\pi}\Gamma((k+1)/2)/\Gamma(k/2+1)$.

- The angle $t$ that the projection of $\mathbf{Z}_2$ into the **x-y** plane makes with the projection of $\mathbf{Z}_1$ then satisfies $\tan(t) = \sin(\varphi_1)\cos(\varphi_2)/(\sin(\phi)\cos(\varphi_1)) = \cos(\varphi_1)/\sin(\phi) \times \tan(\varphi_1)$.

Applying these observations, we get:

$\mathbb{E}[X_i X_j]$

$= \mathbb{E}[(\mathbf{r}_i^\top \mathbf{x})(\mathbf{r}_i^\top \mathbf{y})(\mathbf{r}_j^\top \mathbf{x})(\mathbf{r}_j^\top \mathbf{y})]$

$= \|\mathbf{x}\|_2^2 \|\mathbf{y}\|_2^2 n^2 \int_0^{\pi/2} d\phi f(\phi) \cos^2(\phi) \int_0^\pi d\varphi_1 \sin^{n-3}(\varphi_1) I(n-3)^{-1} \int_0^\pi d\varphi_2 \sin^{n-4}(\varphi_2) I(n-4)^{-1} \times$

$$\int_0^{2\pi} \frac{d\psi}{2\pi} \left( \sin^2(\phi) \cos^2(\varphi_1) + \sin^2(\varphi_1) \cos^2(\varphi_2) \right) \cos(\psi) \cos(\psi + \theta) \cos(t - \psi) \cos(t - \theta - \psi). \tag{21}$$

We first apply the cosine product formula to the two adjacent pairs making up the final product of four cosines involving $\psi$ in the integrand above. The majority of these terms vanish upon integrating with respect to $\psi$, due to the periodicity of the integrands wrt $\psi$. We are thus left with:

$$\mathbb{E}[X_i X_j]$$
$$= \|\mathbf{x}\|_2^2 \|\mathbf{y}\|_2^2 n^2 \int_0^{\pi/2} d\phi f(\phi) \cos^2(\phi) \int_0^\pi d\varphi_1 \sin^{n-3}(\varphi_1) I(n-3)^{-1} \int_0^\pi d\varphi_2 \sin^{n-4}(\varphi_2) I(n-4)^{-1} \times$$
$$\left( \sin^2(\phi) \cos^2(\varphi_1) + \sin^2(\varphi_1) \cos^2(\varphi_2) \right) \left( \frac{1}{4} \cos^2(\theta) + \frac{1}{8} \cos(2t) \right). \tag{22}$$

We now consider two constituent parts of the integral above: one involving the term $\frac{1}{4} \cos^2(\theta)$, and the other involving $\frac{1}{8} \cos(2t)$. We deal first with the former; its evaluation requires several standard trigonometric integrals:

$$\|\mathbf{x}\|_2^2 \|\mathbf{y}\|_2^2 n^2 \int_0^{\pi/2} d\phi f(\phi) \cos^2(\phi) \int_0^\pi d\varphi_1 \sin^{n-3}(\varphi_1) I(n-3)^{-1} \int_0^\pi d\varphi_2 \sin^{n-4}(\varphi_2) I(n-4)^{-1} \times$$
$$\left( \sin^2(\phi) \cos^2(\varphi_1) + \sin^2(\varphi_1) \cos^2(\varphi_2) \right) \frac{1}{4} \cos^2(\theta)$$
$$= \frac{\|\mathbf{x}\|_2^2 \|\mathbf{y}\|_2^2 n^2 \cos^2(\theta)}{4 I(n-3) I(n-4)} \int_0^{\pi/2} d\phi f(\phi) \cos^2(\phi) \int_0^\pi d\varphi_1 \sin^{n-3}(\varphi_1) \times$$
$$\left( \sin^2(\phi) \cos^2(\varphi_1) I(n-4) + \sin^2(\varphi_1) \left( I(n-4) - I(n-2) \right) \right)$$
$$= \frac{\|\mathbf{x}\|_2^2 \|\mathbf{y}\|_2^2 n^2 \cos^2(\theta)}{4 I(n-3) I(n-4)} \int_0^{\pi/2} d\phi (n-2) \sin^{n-3}(\phi) \cos(\phi) \cos^2(\phi) \times$$
$$\left( \sin^2(\phi)(I(n-3) - I(n-1)) I(n-4) + I(n-1) \left( I(n-4) - I(n-2) \right) \right)$$
$$= \frac{\|\mathbf{x}\|_2^2 \|\mathbf{y}\|_2^2 n^2 \cos^2(\theta)}{4 I(n-3) I(n-4)} \left( \left( \frac{1}{n} - \frac{1}{n+2} \right) (I(n-3) - I(n-1)) I(n-4) + \right.$$
$$\left. I(n-1) \left( I(n-4) - I(n-2) \right) \left( \frac{1}{n-2} - \frac{1}{n} \right) \right). \tag{23}$$

We may now turn our attention to the other constituent integral of Equation (22), which involves the term $\cos(2t)$. Recall that from our earlier geometric considerations, we have $\tan(t) = \frac{\cos(\varphi_2)}{\sin(\phi)} \tan(\phi_1)$. An elementary trigonometric calculation using the tan half-angle formula yields:

$$\cos(2t) = \cos\left( 2 \arctan\left( \frac{\cos(\varphi_2)}{\sin(\phi)} \tan(\varphi_1) \right) \right)$$
$$= \frac{1 - \frac{\cos^2(\varphi_2)}{\sin^2(\phi)} \tan^2(\varphi_1)}{\frac{\cos^2(\varphi_2)}{\sin^2(\phi)} \tan^2(\varphi_1) + 1}$$
$$= \frac{\sin^2(\phi) \cos^2(\varphi_1) - \cos^2(\varphi_2) \sin^2(\varphi_1)}{\cos^2(\varphi_2) \sin^2(\varphi_1) + \sin^2(\phi) \cos^2(\varphi_1)}. \tag{24}$$

This observation greatly simplifies the integral from Equation (22) involving the term $\cos(2t)$, as follows:

$$\|\mathbf{x}\|_2^2 \|\mathbf{y}\|_2^2 n^2 \int_0^{\pi/2} d\phi f(\phi) \cos^2(\phi) \int_0^\pi d\varphi_1 \sin^{n-3}(\varphi_1) I(n-3)^{-1} \int_0^\pi d\varphi_2 \sin^{n-4}(\varphi_2) I(n-4)^{-1} \times$$
$$\left( \sin^2(\phi) \cos^2(\varphi_1) + \sin^2(\varphi_1) \cos^2(\varphi_2) \right) \frac{1}{8} \cos(2t)$$
$$= \frac{\|\mathbf{x}\|_2^2 \|\mathbf{y}\|_2^2 n^2}{8 I(n-3) I(n-4)} \int_0^{\pi/2} d\phi f(\phi) \cos^2(\phi) \int_0^\pi d\varphi_1 \sin^{n-3}(\varphi_1) \int_0^\pi d\varphi_2 \sin^{n-4}(\varphi_2) \times$$

$$\left(\sin^2(\phi)\cos^2(\varphi_1) + \sin^2(\varphi_1)\cos^2(\varphi_2)\right)\frac{\sin^2(\phi)\cos^2(\varphi_1) - \cos^2(\varphi_2)\sin^2(\varphi_1)}{\cos^2(\varphi_2)\sin^2(\varphi_1) + \sin^2(\phi)\cos^2(\varphi_1)}$$

$$=\frac{\|\mathbf{x}\|_2^2\|\mathbf{y}\|_2^2 n^2}{8I(n-3)I(n-4)}\int_0^{\pi/2}d\phi f(\phi)\cos^2(\phi)\int_0^\pi d\varphi_1 \sin^{n-3}(\varphi_1)\int_0^\pi d\varphi_2 \sin^{n-4}(\varphi_2)\times$$
$$\left(\sin^2(\phi)\cos^2(\varphi_1) - \cos^2(\varphi_2)\sin^2(\varphi_1)\right).$$
$$(25)$$

But now observe that this integral is exactly of the form dealt with in (23), hence we may immediately identify its value as:

$$\frac{\|\mathbf{x}\|_2^2\|\mathbf{y}\|_2^2 n^2}{8I(n-3)I(n-4)}\left(\left(\frac{1}{n} - \frac{1}{n+2}\right)(I(n-3) - I(n-1))I(n-4)-\right.$$
$$\left. I(n-1)\left(I(n-4) - I(n-2)\right)\left(\frac{1}{n-2} - \frac{1}{n}\right)\right). \qquad (26)$$

Thus substituting our calculations back into Equation (22), we obtain:

$$\mathbb{E}[X_i X_j]$$
$$=\frac{\|\mathbf{x}\|_2^2\|\mathbf{y}\|_2^2 n^2}{4I(n-3)I(n-4)}\left(\left(\frac{1}{n} - \frac{1}{n+2}\right)(I(n-3) - I(n-1))I(n-4)\left[\cos^2(\theta) + \frac{1}{2}\right]+\right.$$
$$\left. I(n-1)\left(I(n-4) - I(n-2)\right)\left(\frac{1}{n-2} - \frac{1}{n}\right)\left[\cos^2(\theta) - \frac{1}{2}\right]\right). \quad (27)$$

The covariance term is obtained by subtracting off $\mathbb{E}[X_i]\mathbb{E}[X_i] = \langle\mathbf{x},\mathbf{y}\rangle^2$. Now we sum over $m(m-1)$ covariance terms and take into account the normalization factor $\frac{1}{\sqrt{m}}$ for the Gaussian matrix entries. That gives the extra multiplicative term $\frac{m(m-1)}{m^2} = \frac{m-1}{m}$. Thus we obtain the quantity in the statement of the theorem, completing the proof. □

## 8.3 Proof of Theorem 3.3

We obtain Theorem 3.3 through a sequence of smaller propositions. Broadly, the strategy is first to show that the estimators of Theorem 3.3 are unbiased (Proposition 8.1). An expression for the mean squared error of the estimator $\widehat{K}_m^{(1)}$ with one matrix block is then derived (Proposition 8.2). Finally, a straightforward recursive formula for the mean squared error of the general estimator is derived (Proposition 8.3), and the result of the theorem then follows.

**Proposition 8.1.** *The estimator* $\widehat{K}_m^{(k)}(\mathbf{x},\mathbf{y})$ *is unbiased, for all* $k, n \in \mathbb{N}$, $m \le n$, *and* $\mathbf{x}, \mathbf{y} \in \mathbb{R}^n$.

*Proof.* Notice first that since rows of $\mathbf{S} = \{s_{i,j}\}$ are orthogonal and are $L_2$-normalized, the matrix $\mathbf{S}$ is an isometry. Thus each block $\mathbf{SD}_i$ is also an isometry. Therefore it suffices to prove the claim for $k = 1$.

Then, denoting by $\mathbf{J} = (J_1, \dots, J_m)$ the indices of the randomly selected rows of $\mathbf{SD}_1$, note that the estimator $\widehat{K}_m^{(1)}(\mathbf{x},\mathbf{y})$ may be expressed in the form

$$\widehat{K}_m^{(1)}(\mathbf{x},\mathbf{y}) = \frac{1}{m}\sum_{i=1}^m \left(\sqrt{n}(\mathbf{SD}_1)_{J_i}\mathbf{x} \times \sqrt{n}(\mathbf{SD}_1)_{J_i}\mathbf{y}\right),$$

where $(\mathbf{SD}_1)_i$ is the $i^{\text{th}}$ row of $\mathbf{SD}_1$. Since each of the rows of $\mathbf{SD}_1$ has the same marginal distribution, it suffices to demonstrate that $\mathbb{E}[\mathbf{y}^T\mathbf{D}_1\mathbf{S}_1^\top\mathbf{S}_1\mathbf{D}_1\mathbf{x}] = \frac{\mathbf{x}^\top\mathbf{y}}{n}$, where $\mathbf{S}_1$ is the first row of $\mathbf{S}$. Now note

$$\mathbb{E}[\mathbf{y}^\top\mathbf{DS}_1^\top\mathbf{S}_1\mathbf{Dx}] = \frac{1}{n}\mathbb{E}\left[\sum_{i=1}^n y_i d_i \times \sum_{i=1}^n x_i d_i\right] = \frac{1}{n}\mathbb{E}\left[\sum_{i=1}^n x_i y_i d_i^2\right] + \mathbb{E}\left[\sum_{i\ne j} x_i y_j d_i d_j\right] = \frac{\mathbf{x}^\top\mathbf{y}}{n},$$

where $d_i = \mathbf{D}_{ii}$ are iid Rademacher random variables, for $i = 1, \dots, n$. □

With Proposition 8.1 in place, the mean square error for the estimator $\widehat{K}_m^{(1)}$ using one matrix block can be derived.

**Proposition 8.2.** *The MSE of the single* $\mathbf{SD}^{(\mathcal{R})}$*-block $m$-feature estimator $\widehat{K}_m^{(1)}(\mathbf{x}, \mathbf{y})$ for $\langle \mathbf{x}, \mathbf{y} \rangle$ using the without replacement row sub-sampling strategy is*

$$\mathrm{MSE}(\widehat{K}_m^{(1)}(\mathbf{x}, \mathbf{y})) = \frac{1}{m}\left(\frac{n-m}{n-1}\right)\left(\|\mathbf{x}\|^2\|\mathbf{y}\|^2 + \langle \mathbf{x}, \mathbf{y} \rangle^2 - 2\sum_{i=1}^{n} x_i^2 y_i^2\right).$$

*Proof.* First note that since $\widehat{K}_m^{(1)}(\mathbf{x}, \mathbf{y})$ is unbiased, the mean squared error is simply the variance of this estimator. Secondly, denoting the indices of the $m$ randomly selected rows by $\mathbf{J} = (J_1, \ldots, J_m)$, by conditioning on $\mathbf{J}$ we obtain the following:

$$\mathrm{Var}\left(\widehat{K}_m^{(1)}(\mathbf{x}, \mathbf{y})\right) =$$

$$\frac{n^2}{m^2}\left(\mathbb{E}\left[\mathrm{Var}\left(\sum_{p=1}^{m}(\mathbf{SDx})_{J_p}(\mathbf{SDy})_{J_p}\middle|\mathbf{J}\right)\right] + \mathrm{Var}\left(\mathbb{E}\left[\sum_{p=1}^{m}(\mathbf{SDx})_{J_p}(\mathbf{SDy})_{J_p}\middle|\mathbf{J}\right]\right)\right).$$

Now note that the conditional expectation in the second term is constant as a function of $J$, since conditional on whichever rows are sampled, the resulting estimator is unbiased. Taking the variance of this constant therefore causes the second term to vanish. Now consider the conditional variance that appears in the first term:

$$\mathrm{Var}\left(\sum_{p=1}^{m}(\mathbf{SDx})_{J_p}(\mathbf{SDy})_{J_p}\middle|\mathbf{J}\right) = \sum_{p=1}^{m}\sum_{p'=1}^{m}\mathrm{Cov}\left((\mathbf{SDx})_{J_m}(\mathbf{SDy})_{J_p}, (\mathbf{SDx})_{J_{p'}}(\mathbf{SDy})_{J_{p'}}\middle|\mathbf{J}\right)$$

$$= \sum_{p,p'=1}^{m}\sum_{i,j,k,l=1}^{n} s_{J_p i}s_{J_p j}s_{J_{p'}k}s_{J_{p'}l}x_i y_j x_k y_l\mathrm{Cov}\left(d_i d_j, d_k d_l\right),$$

where we write $\mathbf{D} = \mathrm{Diag}(d_1, \ldots, d_n)$. Now note that $\mathrm{Cov}\left(d_i d_j, d_k d_l\right)$ is non-zero iff $i, j$ are distinct, and $\{i, j\} = \{k, l\}$, in which case the covariance is 1. We therefore obtain:

$$\mathrm{Var}\left(\sum_{p=1}^{m}(\mathbf{SDx})_{J_p}(\mathbf{SDy})_{J_p}\middle|\mathbf{J}\right) =$$

$$\sum_{p,p'=1}^{m}\sum_{i\neq j}^{n}\left(s_{J_p i}s_{J_p j}s_{J_{p'} i}s_{J_{p'} j}x_i^2 y_j^2 + s_{J_p i}s_{J_p j}s_{J_{p'} j}s_{J_{p'} i}x_i y_j x_j y_i\right).$$

Substituting this expression for the conditional variance into the decomposition of the MSE of the estimator, we obtain the result of the theorem:

$$\mathrm{Var}\left(\widehat{K}_m^{(1)}(\mathbf{x}, \mathbf{y})\right) = \frac{n^2}{m^2}\mathbb{E}\left[\sum_{p,p'=1}^{m}\sum_{i\neq j}^{n}\left(s_{J_p i}s_{J_p j}s_{J_{p'} i}s_{J_{p'} j}x_i^2 y_j^2 + s_{J_p i}s_{J_p j}s_{J_{p'} j}s_{J_{p'} i}x_i y_j x_j y_i\right)\right]$$

$$= \frac{n^2}{m^2}\sum_{p,p'=1}^{m}\sum_{i\neq j}^{n}\left(x_i^2 y_j^2 + x_i x_j y_i y_j\right)\mathbb{E}\left[s_{J_p i}s_{J_p j}s_{J_{p'} i}s_{J_{p'} j}\right].$$

We now consider the law on the index variables $\mathbf{J} = (J_1, \ldots, J_m)$ induced by the sub-sampling strategy without replacement to evaluate the expectation in this last term. If $p = p'$, the integrand of the expectation is deterministically $1/n^2$. If $p \neq p'$, then we obtain:

$$\mathbb{E}\left[s_{J_p i}s_{J_p j}s_{J_{p'} i}s_{J_{p'} j}\right] = \mathbb{E}\left[s_{J_p i}s_{J_p j}\mathbb{E}\left[s_{J_{p'} i}s_{J_{p'} j}\middle|J_p\right]\right]$$

$$= \mathbb{E}\left[s_{J_p i}s_{J_p j}\left[\left(\frac{1}{n}\left(\frac{n/2-1}{n-1}\right) - \frac{1}{n}\left(\frac{n/2}{n-1}\right)\right)\mathbb{1}_{\{s_{J_p i}s_{J_p j}=1/n\}} + \left(\frac{1}{n}\left(\frac{n/2}{n-1}\right) - \frac{1}{n}\left(\frac{n/2-1}{n-1}\right)\right)\mathbb{1}_{\{s_{J_p i}s_{J_p j}=-1/n\}}\right]\right]$$

$$= \frac{1}{n(n-1)} \mathbb{E}\left[ s_{J_p i} s_{J_p j} \left( \mathbb{1}_{\{s_{J_p i} s_{J_p j} = -1/n\}} - \mathbb{1}_{\{s_{J_p i} s_{J_p j} = 1/n\}} \right) \right]$$

$$= \frac{1}{n^2(n-1)},$$

where we have used the fact that the products $s_{J_p i} s_{J_p j}$ and $s_{J_{p'} i} s_{J_{p'} j}$ take values in $\{\pm 1/n\}$, and because distinct rows of $\mathbf{S}$ are orthogonal, the marginal probability of each of the two values is $1/2$. A simple adjustment, using almost-sure distinctness of $J_p$ and $J_{p'}$, yields the conditional probabilities needed to evaluate the conditional expectation that appears in the calculation above.

Substituting the values of these expectations back into the expression for the variance of $\widehat{K}_m^{(1)}(\mathbf{x}, \mathbf{y})$ then yields

$$
\begin{aligned}
\mathrm{Var}(\widehat{K}_m^{(1)}(\mathbf{x}, \mathbf{y})) &= \frac{n^2}{m^2} \sum_{i \neq j}^{n} \left( x_i^2 y_j^2 + x_i x_j y_i y_j \right) \left( m \times \frac{1}{n^2} - m(m-1) \times \frac{1}{n^2(n-1)} \right) \\
&= \frac{1}{m} \left( 1 - \frac{m-1}{n-1} \right) \sum_{i \neq j}^{n} \left( x_i^2 y_j^2 + x_i x_j y_i y_j \right) \\
&= \frac{1}{m} \left( 1 - \frac{m-1}{n-1} \right) \left( \sum_{i,j=1}^{n} \left( x_i^2 y_j^2 + x_i x_j y_i y_j \right) - 2 \sum_{i=1}^{n} x_i^2 y_i^2 \right) \\
&= \frac{1}{m} \left( \frac{n-m}{n-1} \right) \left( \langle \mathbf{x}, \mathbf{y} \rangle^2 + \|\mathbf{x}\|^2 \|\mathbf{y}\|^2 - 2 \sum_{i=1}^{n} x_i^2 y_i^2 \right),
\end{aligned}
$$

as required.

$\square$

We now turn our attention to the following recursive expression for the mean squared error of a general estimator.

**Proposition 8.3.** *Let $k \geq 2$. We have the following recursion for the MSE of $K_m^{(k)}(x, y)$:*

$$\mathrm{MSE}(\widehat{K}_m^{(k)}(\mathbf{x}, \mathbf{y})) = \mathbb{E}\left[ \mathrm{MSE}\left( \widehat{K}_m^{(k-1)}(\mathbf{SD}_1 \mathbf{x}, \mathbf{SD}_1 \mathbf{y}) | \mathbf{D}_1 \right) \right].$$

*Proof.* The result follows from a straightforward application of the law of total variance, conditioning on the matrix $\mathbf{D}_1$. Observe that

$$
\begin{aligned}
\mathrm{MSE}(\widehat{K}_m^{(k)}(\mathbf{x}, \mathbf{y})) &= \mathrm{Var}(\widehat{K}_m^{(k)}(\mathbf{x}, \mathbf{y})) \\
&= \mathbb{E}\left[ \mathrm{Var}\left( \widehat{K}_m^{(k)}(\mathbf{x}, \mathbf{y}) \middle| \mathbf{D}_1 \right) \right] + \mathrm{Var}\left( \mathbb{E}\left[ \widehat{K}_m^{(k)}(\mathbf{x}, \mathbf{y}) \middle| \mathbf{D}_1 \right] \right) \\
&= \mathbb{E}\left[ \mathrm{Var}\left( \widehat{K}_m^{(k-1)}(\mathbf{SD}_1 \mathbf{x}, \mathbf{SD}_1 \mathbf{y}) \middle| \mathbf{D}_1 \right) \right] + \mathrm{Var}\left( \mathbb{E}\left[ \widehat{K}_m^{(k-1)}(\mathbf{SD}_1 \mathbf{x}, \mathbf{SD}_1 \mathbf{y}) \middle| \mathbf{D}_1 \right] \right).
\end{aligned}
$$

But examining the conditional expectation in the second term, we observe

$$\mathbb{E}\left[ \widehat{K}_m^{(k-1)}(\mathbf{SD}_1 \mathbf{x}, \mathbf{SD}_1 \mathbf{y}) \middle| \mathbf{D}_1 \right] = \langle \mathbf{SD}_1 \mathbf{x}, \mathbf{SD}_1 \mathbf{y} \rangle \quad \text{almost surely},$$

by unbiasedness of the estimator, and since $\mathbf{SD}_1$ is orthogonal almost surely, this is equal to the (constant) inner product $\langle \mathbf{x}, \mathbf{y} \rangle$ almost surely. This conditional expectation therefore has $0$ variance, and so the second term in the expression for the MSE above vanishes, which results in the statement of the proposition. $\square$

With these intermediate propositions established, we are now in a position to prove Theorem 3.3. In order to use the recursive result of Proposition 8.3, we require the following lemma.

**Lemma 8.4.** *For all $x, y, \in \mathbb{R}^n$, we have*

$$\mathbb{E}\left[ \sum_{i=1}^{n} (\mathbf{SDx})_i^2 (\mathbf{SDy})_i^2 \right] = \frac{1}{n} \left( \|\mathbf{x}\|^2 \|\mathbf{y}\|^2 + 2\langle \mathbf{x}, \mathbf{y} \rangle^2 - 2 \sum_{i=1}^{n} x_i^2 y_i^2 \right).$$

*Proof.* The result follows by direct calculation. Note that

$$\mathbb{E}\left[\sum_{i=1}^{n}(\mathbf{SD}\mathbf{x})_{i}^{2}(\mathbf{SD}\mathbf{y})_{i}^{2}\right]=n\mathbb{E}\left[\left(\sum_{a=1}^{n}s_{1a}d_{a}x_{a}\right)^{2}\left(\sum_{a=1}^{n}s_{1a}d_{a}y_{a}\right)^{2}\right]$$

$$=n\sum_{i,j,k,l=1}^{n}s_{1i}s_{1j}s_{1k}s_{1l}x_{i}x_{j}y_{k}y_{l}\mathbb{E}\left[d_{i}d_{j}d_{k}d_{l}\right],$$

where the first inequality follows since the $n$ summands indexed by $i$ in the initial expectation are identically distributed. Now note that the expectation $\mathbb{E}\left[d_{i}d_{j}d_{k}d_{l}\right]$ is non-zero iff $i=j=k=l$, or $i=j\neq k=l$, or $i=k\neq j=l$, or $i=l\neq k=j$; in all such cases, the expectation takes the value 1. Substituting this into the above expression and collecting terms, we obtain

$$\mathbb{E}\left[\sum_{i=1}^{n}(\mathbf{SD}\mathbf{x})_{i}^{2}(\mathbf{SD}\mathbf{y})_{i}^{2}\right]=\frac{1}{n}\left(\sum_{i=1}^{n}x_{i}^{2}y_{i}^{2}+\sum_{i\neq j}x_{i}^{2}y_{j}^{2}+2\sum_{i\neq j}x_{i}x_{j}y_{i}y_{j}\right)$$

$$=\frac{1}{n}\left(\sum_{i,j=1}^{n}x_{i}^{2}y_{j}^{2}+2\sum_{i,j=1}^{n}x_{i}x_{j}y_{i}y_{j}-2\sum_{i=1}^{n}x_{i}^{2}y_{i}^{2}\right),$$

from which the statement of the lemma follows immediately. $\square$

*Proof of Theorem 3.3.* Recall that we aim to establish the following general expression for $k\geq 1$:

$$\mathrm{MSE}(\widehat{K}_{m}^{(k)}(\mathbf{x},\mathbf{y}))=$$
$$\frac{1}{m}\left(\frac{n-m}{n-1}\right)\left(((\mathbf{x}^{\top}\mathbf{y})^{2}+\|\mathbf{x}\|^{2}\|\mathbf{y}\|^{2})+\sum_{r=1}^{k-1}\frac{(-1)^{r}2^{r}}{n^{r}}(2(\mathbf{x}^{\top}\mathbf{y})^{2}+\|\mathbf{x}\|^{2}\|\mathbf{y}\|^{2})+\frac{(-1)^{k}2^{k}}{n^{k-1}}\sum_{i=1}^{n}x_{i}^{2}y_{i}^{2}\right).$$

We proceed by induction. The case $k=1$ is verified by Proposition 8.2. For the inductive step, suppose the result holds for some $k\in\mathbb{N}$. Then observe by Proposition 8.3 and the induction hypothesis, we have

$$\mathrm{MSE}(\widehat{K}_{m}^{(k+1)}(\mathbf{x},\mathbf{y}))=\mathbb{E}\left[\mathrm{MSE}\left(\widehat{K}_{m}^{(k-1)}(\mathbf{SD}_{1}\mathbf{x},\mathbf{SD}_{1}\mathbf{y})|\mathbf{D}_{1}\right)\right]$$

$$=\frac{1}{m}\left(\frac{n-m}{n-1}\right)\left(((\mathbf{x}^{\top}\mathbf{y})^{2}+\|\mathbf{x}\|^{2}\|\mathbf{y}\|^{2})+\sum_{r=1}^{k-1}\frac{(-1)^{r}2^{r}}{n^{r}}(2(\mathbf{x}^{\top}\mathbf{y})^{2}+\|\mathbf{x}\|^{2}\|\mathbf{y}\|^{2})\right.$$

$$\left.+\frac{(-1)^{k}2^{k}}{n^{k-1}}\sum_{i=1}^{n}\mathbb{E}\left[(\mathbf{SD}_{1}\mathbf{x})_{i}^{2}(\mathbf{SD}_{1}\mathbf{y})_{i}^{2}\right]\right),$$

where we have used that $\mathbf{SD}_{1}$ is almost surely orthogonal, and therefore $\|\mathbf{SD}_{1}\mathbf{x}\|^{2}=\|\mathbf{x}\|^{2}$ almost surely, $\|\mathbf{SD}_{1}\mathbf{y}\|^{2}=\|\mathbf{y}\|^{2}$ almost surely, and $\langle\mathbf{SD}_{1}\mathbf{x},\mathbf{SD}_{1}\mathbf{y}\rangle=\langle\mathbf{x},\mathbf{y}\rangle$ almost surely. Applying Lemma 8.4 to the remaining expectation and collecting terms yields the required expression for $\mathrm{MSE}(\widehat{K}_{m}^{(k+1)}(\mathbf{x},\mathbf{y}))$, and the proof is complete. $\square$

## 8.4 Proof of Lemma 3.5

*Proof.* Consider the last block $\mathbf{H}$ that is sub-sampled. Notice that if rows $\mathbf{r}^{1}$ and $\mathbf{r}^{2}$ of $\mathbf{H}$ of indices $i$ and $\frac{n}{2}+i$ are chosen then from the recursive definition of $\mathbf{H}$ we conclude that $(\mathbf{r}^{2})^{\top}\mathbf{x}=(\mathbf{r}_{1}^{1})^{\top}\mathbf{x}-(\mathbf{r}_{2}^{1})^{\top}\mathbf{x}$, where $\mathbf{r}_{1}^{1},\mathbf{r}_{2}^{1}$ stand for the first and second half of $\mathbf{r}^{1}$ respectively. Thus computations of $(\mathbf{r}^{1})^{\top}\mathbf{x}$ can be reused to compute both $(\mathbf{r}^{1})^{\top}\mathbf{x}$ and $(\mathbf{r}^{2})^{\top}\mathbf{x}$ in time $n+O(1)$ instead of $2n$. If we denote by $r$ the expected number of pairs of rows $(i,\frac{n}{2}+i)$ that are chosen by the random sampling mechanism, then we see that by applying the trick above for all the $r$ pairs, we obtain time complexity $O((k-1)n\log(n)+n(m-2r)+nr+r)$, where: $O((k-1)n\log(n))$ is the time required to compute first $(k-1)$ $\mathbf{HD}$ blocks (with the use of Walsh-Hadamard Transform), $O(n(m-2r))$ stands for time complexity of the brute force computations for these rows that were not coupled in the last block and $O(nr+r)$ comes from the above trick applied to all $r$ aforementioned pairs of

rows. Thus, to obtain the first term in the min-expression on time complexity from the statement of the lemma, it remains to show that

$$\mathbb{E}[r] = \frac{(m-1)m}{2(n-1)}. \tag{28}$$

But this is straightforward. Note that the number of the $m$-subsets of the set of all $n$ rows that contain some fixed rows of indices $i_1$, $i_2$ ($i_1 \neq i_2$) is $\binom{n-2}{m-2}$. Thus for any fixed pair of rows of indices $i$ and $\frac{n}{2} + i$ the probability that these two rows will be selected is exactly $p_{succ} = \frac{\binom{n-2}{m-2}}{\binom{n}{m}} = \frac{(m-1)m}{(n-1)n}$. Equation 28 comes from the fact that clearly: $\mathbb{E}[r] = \frac{n}{2}p_{succ}$. Thus we obtain the first term in the min-expression from the statement of the lemma. The other one comes from the fact that one can always do all the computations by calculating $k$ times Walsh-Hadamard transformation. That completes the proof.

$\square$

## 8.5  Proof of Theorem 3.6

The proof of Theorem 3.6 follows a very similar structure to that of Theorem 3.3; we proceed by induction, and may use the results of Proposition 8.3 to set up a recursion. We first show unbiasedness of the estimator (Proposition 8.5), and then treat the base case of the inductive argument (Proposition 8.6). We prove slightly more general statements than needed for Theorem 3.6, as this will allow us to explore the fully complex case in §8.7.

**Proposition 8.5.** *The estimator $K_m^{\mathcal{H},(k)}(\mathbf{x}, \mathbf{y})$ is unbiased for all $k, n \in \mathbb{N}$, $m \leq n$, and $\mathbf{x}, \mathbf{y} \in \mathbb{C}^n$ with $\langle \overline{\mathbf{x}}, \mathbf{y} \rangle \in \mathbb{R}$; in particular, for all $\mathbf{x}, \mathbf{y} \in \mathbb{R}$.*

*Proof.* Following a similar argument to the proof of Proposition 8.1, note that it is sufficient to prove the claim for $k = 1$, since each $\mathbf{SD}$ block is unitary, and hence preserves the Hermitian product $\langle \overline{\mathbf{x}}, \mathbf{y} \rangle$.

Next, note that the estimator can be written as a sum of identically distributed terms:

$$\widehat{K}_m^{\mathcal{H},(1)}(\mathbf{x}, \mathbf{y}) = \frac{n}{m} \sum_{i=1}^{m} \mathrm{Re}\left( (\mathbf{S}\overline{\mathbf{D}}_1 \overline{\mathbf{x}})_{J_i} \times (\mathbf{SD}_1 \mathbf{y})_{J_i} \right) .$$

The terms are identically distributed since the index variables $J_i$ are marginally identically distributed, and the rows of $\mathbf{SD}_1$ are marginally identically distributed (the elements of a row are iid $\mathrm{Unif}(S^1)/\sqrt{n}$). Now note

$$\mathbb{E}\left[ \mathrm{Re}\left( (\mathbf{S}\overline{\mathbf{D}}_1 \overline{\mathbf{x}})_{J_i} \times (\mathbf{SD}_1 \mathbf{y})_{J_i} \right) \right] = \frac{1}{n}\mathbb{E}\left[ \sum_{i=1}^{n} y_i d_i \times \sum_{i=1}^{n} \overline{x}_i \overline{d}_i \right]$$

$$= \frac{1}{n}\mathbb{E}\left[ \sum_{i=1}^{n} \overline{x}_i y_i d_i \overline{d}_i \right] + \mathbb{E}\left[ \sum_{i \neq j} \overline{x}_i y_j \overline{d}_i d_j \right] = \frac{1}{n}\langle \overline{\mathbf{x}}, \mathbf{y} \rangle ,$$

where $d_i = \mathbf{D}_{ii} \overset{iid}{\sim} \mathrm{Unif}(S^1)$ for $i = 1, \dots, n$. This immediately yields $\mathbb{E}\left[ \widehat{K}_m^{\mathcal{H},(1)}(\mathbf{x}, \mathbf{y}) \right] = \langle \overline{\mathbf{x}}, \mathbf{y} \rangle$, as required. $\square$

We now derive the base case for our inductive proof, again proving a slightly more general statement then necessary for Theorem 3.6.

**Proposition 8.6.** *Let $\mathbf{x}, \mathbf{y} \in \mathbb{C}^n$ such that $\langle \overline{\mathbf{x}}, \mathbf{y} \rangle \in \mathbb{R}$. The MSE of the single complex $\mathbf{SD}$-block $m$-feature estimator $K_m^{\mathcal{H},(1)}(\mathbf{x}, \mathbf{y})$ for $\langle \overline{\mathbf{x}}, \mathbf{y} \rangle$ is*

$$\mathrm{MSE}(\widehat{K}_m^{\mathcal{H},(1)}(\mathbf{x}, \mathbf{y})) = \frac{1}{2m}\left( \frac{n-m}{n-1} \right) \left( \langle \overline{\mathbf{x}}, \mathbf{x} \rangle \langle \overline{\mathbf{y}}, \mathbf{y} \rangle + \langle \overline{\mathbf{x}}, \mathbf{y} \rangle^2 - \sum_{r=1}^{n} |x_r|^2 |y_r|^2 - \sum_{r=1}^{n} \mathrm{Re}(\overline{x}_r^2 y_r^2) \right) .$$

*Proof.* The proof is very similar to that of Proposition 8.2. By the unbiasedness result of Proposition 8.5, the mean squared error of the estimator is simply the variance. We begin by conditioning on the random index vector $\mathbf{J}$ selected by the sub-sampling procedure.

$$\widehat{K}_m^{\mathcal{H},(1)}(\mathbf{x},\mathbf{y})) = \frac{1}{M}\mathrm{Re}\left(\langle\sqrt{n}(\mathbf{S}\overline{\mathbf{D}}_1\overline{\mathbf{x}})_{\mathbf{J}},\,\sqrt{n}(\mathbf{S}\mathbf{D}\mathbf{y})_{\mathbf{J}}\rangle\right),$$

where again $\mathbf{J}$ is a set of uniform iid indices from $1,\ldots,n$, and the bar over $D$ represents complex conjugation. Since the estimator is again unbiased, its MSE is equal to its variance. First conditioning on the index set $\mathbf{J}$, as for Proposition 8.6, we obtain

$$\mathrm{Var}\left(\widehat{K}_m^{\mathcal{H},(1)}(x,y)\right)$$
$$=\frac{n^2}{m^2}\left(\mathbb{E}\left[\mathrm{Var}\left(\mathrm{Re}\left(\sum_{p=1}^m(\mathbf{S}\overline{\mathbf{D}}_1\overline{\mathbf{x}})_{J_p}(\mathbf{S}\mathbf{D}_1\mathbf{y})_{J_p}\right)\middle|\mathbf{J}\right)\right]+\mathrm{Var}\left(\mathbb{E}\left[\mathrm{Re}\left(\sum_{p=1}^m(\mathbf{S}\overline{\mathbf{D}}_1\overline{\mathbf{x}})_{J_p}(\mathbf{S}\mathbf{D}_1\mathbf{y})_{J_p}\right)\middle|\mathbf{J}\right]\right)\right).$$

Again, the second term vanishes as the conditional expectation is constant as a function of $\mathbf{J}$, by unitarity of $\mathbf{S}\mathbf{D}$. Turning attention to the conditional variance expression in the first term, we note

$$\mathrm{Var}\left(\mathrm{Re}\left(\sum_{p=1}^m(\mathbf{S}\overline{\mathbf{D}}_1\overline{\mathbf{x}})_{J_p}(\mathbf{S}\mathbf{D}_1\mathbf{y})_{J_p}\right)\middle|\mathbf{J}\right) =$$
$$\sum_{p,p'=1}^m\sum_{i,j,k,l=1}^n s_{J_pi}s_{J_pj}s_{J_{p'}k}s_{J_{p'}l}\mathrm{Cov}\left(\mathrm{Re}(\overline{d}_i\overline{x}_id_jy_j),\mathrm{Re}(\overline{d}_k\overline{x}_kd_ly_l)\right).$$

Now note that the covariance term is non-zero iff $i,j$ are distinct, and $\{i,j\}=\{k,l\}$. We therefore obtain

$$\mathrm{Var}\left(\mathrm{Re}\left(\sum_{p=1}^m(\mathbf{S}\overline{\mathbf{D}}\overline{\mathbf{x}})_{J_p}(\mathbf{S}\mathbf{D}\mathbf{y})_{J_p}\right)\middle|\mathbf{J}\right)$$
$$=\sum_{p,p'=1}^m\sum_{i\neq j}^n s_{J_pi}s_{J_pj}s_{J_{p'}i}s_{J_{p'}j}\left(\mathrm{Cov}\left(\mathrm{Re}(\overline{d}_i\overline{x}_id_jy_j),\mathrm{Re}(\overline{d}_i\overline{x}_id_jy_j)\right)+\mathrm{Cov}\left(\mathrm{Re}(\overline{d}_i\overline{x}_id_jy_j),\mathrm{Re}(\overline{d}_j\overline{x}_jd_iy_i)\right)\right)$$

First consider the term $\mathrm{Cov}\left(\mathrm{Re}(\overline{d}_i\overline{x}_id_jy_j),\mathrm{Re}(\overline{d}_i\overline{x}_id_jy_j)\right)$. The random variable $\overline{d}_i\overline{x}_id_jy_j$ is distributed uniformly on the circle in the complex plane centered at the origin with radius $|\overline{x}_iy_j|$. Therefore the variance of its real part is

$$\mathrm{Cov}\left(\mathrm{Re}(\overline{d}_i\overline{x}_id_jy_j),\mathrm{Re}(\overline{d}_i\overline{x}_id_jy_j)\right)=\frac{1}{2}|\overline{x}_iy_j|^2=\frac{1}{2}x_i\overline{x}_iy_j\overline{y}_j.$$

For the second covariance term, we perform an explicit calculation. Let $Z=e^{i\theta}=\overline{d}_id_j$. Then we have

$$\mathrm{Cov}\left(\mathrm{Re}(\overline{d}_i\overline{x}_id_jy_j),\mathrm{Re}(\overline{d}_j\overline{x}_jd_iy_i)\right)=\mathrm{Cov}\left(\mathrm{Re}(Z\overline{x}_iy_j),\mathrm{Re}(\overline{Z}\overline{x}_jy_i)\right)$$
$$=\mathrm{Cov}\left(\cos(\theta)\mathrm{Re}(\overline{x}_iy_j)-\sin(\theta)\mathrm{Im}(\overline{x}_iy_j),\cos(\theta)\mathrm{Re}(\overline{x}_jy_i)+\sin(\theta)\mathrm{Im}(\overline{x}_jy_i)\right)$$
$$=\frac{1}{2}\left(\mathrm{Re}(\overline{x}_iy_j)\mathrm{Re}(\overline{x}_jy_i)-\mathrm{Im}(\overline{x}_iy_j)\mathrm{Im}(\overline{x}_jy_i)\right),$$

with the final equality following since the angle $\theta$ is uniformly distributed on $[0,2\pi]$, and standard trigonometric integral identities. We recognize the bracketed terms in the final line as the real part of the product $\overline{x}_i\overline{x}_jy_iy_j$. Substituting these into the expression for the conditional variance obtained above, we have

$$\mathrm{Var}\left(\mathrm{Re}\left(\sum_{p=1}^m(\mathbf{S}\overline{\mathbf{D}}\mathbf{x})_{J_p}(\mathbf{S}\mathbf{D}\mathbf{y})_{J_p}\right)\middle|\mathbf{J}\right)=\sum_{p,p'=1}^m\sum_{i\neq j}^n s_{J_pi}s_{J_pj}s_{J_{p'}i}s_{J_{p'}j}\frac{1}{2}\left(x_i\overline{x}_iy_j\overline{y}_j+\mathrm{Re}(\overline{x}_i\overline{x}_jy_iy_j)\right).$$

Now taking the expectation over the index variables $\mathbf{J}$, we note that as in the proof of Proposition 8.2, the expectation of the term $s_{J_pi}s_{J_pj}s_{J_{p'}i}s_{J_{p'}j}$ is $1/n^2$ when $p=p'$, and $1/(n^2(n-1))$ otherwise. Therefore we obtain

$$\mathrm{Var}\left(\widehat{K}_m^{\mathcal{H},(1)}(\mathbf{x},\mathbf{y})\right)=\frac{n^2}{m^2}\left(\left(\frac{m}{n^2}+\frac{m(m-1)}{n^2(n-1)}\right)\frac{1}{2}\sum_{i\neq j}^n\left(x_i\overline{x}_iy_j\overline{y}_j+\mathrm{Re}(\overline{x}_i\overline{x}_jy_iy_j)\right)\right)$$

$$= \frac{1}{2m}\left(\frac{n-m}{n-1}\right)\left(\sum_{i \neq j}^{n} \left(x_i\overline{x}_iy_j\overline{y}_j + \operatorname{Re}(\overline{x}_i\overline{x}_jy_iy_j)\right)\right)$$

$$= \frac{1}{2m}\left(\frac{n-m}{n-1}\right)\left(\sum_{i,j=1}^{n} \left(x_i\overline{x}_iy_j\overline{y}_j + \operatorname{Re}(\overline{x}_i\overline{x}_jy_iy_j)\right) - \sum_{i=1}^{n}(x_i\overline{x}_iy_i\overline{y}_i + \operatorname{Re}(\overline{x}_i\overline{x}_iy_iy_i))\right)$$

$$= \frac{1}{2m}\left(\frac{n-m}{n-1}\right)\left(\langle\overline{\mathbf{x}}, \mathbf{x}\rangle\langle\overline{\mathbf{y}}, \mathbf{y}\rangle + \langle\overline{\mathbf{x}}, \mathbf{y}\rangle^2 - \sum_{i=1}^{n}(x_i\overline{x}_iy_i\overline{y}_i + \operatorname{Re}(\overline{x}_i\overline{x}_iy_iy_i))\right),$$

where in the final equality we have used the assumption that $\langle\overline{\mathbf{x}}, \mathbf{y}\rangle \in \mathbb{R}$. $\qquad\square$

We are now in a position to prove Theorem 3.6 by induction, using Proposition 8.6 as a base case, and Proposition 8.3 for the inductive step.

*Proof of Theorem 3.6.* Recall that we aim to establish the following general expression for $k \geq 1$:

$$\operatorname{MSE}(\widehat{K}_m^{\mathcal{H},(k)}(\mathbf{x}, \mathbf{y})) = \frac{1}{2m}\left(\frac{n-m}{n-1}\right)\left(((\mathbf{x}^\top\mathbf{y})^2 + \|\mathbf{x}\|^2\|\mathbf{y}\|^2) + \right.$$

$$\left.\sum_{r=1}^{k-1}\frac{(-1)^r 2^r}{n^r}(2(\mathbf{x}^\top\mathbf{y})^2 + \|\mathbf{x}\|^2\|\mathbf{y}\|^2) + \frac{(-1)^k 2^k}{n^{k-1}}\sum_{i=1}^{n}x_i^2 y_i^2\right).$$

We proceed by induction. The case $k = 1$ is verified by Proposition 8.6, and by noting that in the expression obtained in Proposition 8.6, we have

$$\sum_{i=1}^{n} x_i\overline{x}_iy_i\overline{y}_i = \operatorname{Re}(\overline{x}_i\overline{x}_iy_iy_i) = \sum_{i=1}^{n} x_i^2 y_i^2.$$

For the inductive step, suppose the result holds for some $k \in \mathbb{N}$. Then observe by Proposition 8.3 and the induction hypothesis, we have, for $\mathbf{x}, \mathbf{y} \in \mathbb{R}^n$:

$$\operatorname{MSE}(\widehat{K}_m^{\mathcal{H},(k+1)}(\mathbf{x}, \mathbf{y})) = \mathbb{E}\left[\operatorname{MSE}\left(\widehat{K}_m^{(k-1)}(\mathbf{SD}_1\mathbf{x}, \mathbf{SD}_1\mathbf{y})|\mathbf{D}_1\right)\right]$$

$$= \frac{1}{2m}\left(\frac{n-m}{n-1}\right)\left(((\mathbf{x}^\top\mathbf{y})^2 + \|\mathbf{x}\|^2\|\mathbf{y}\|^2) + \sum_{r=1}^{k-1}\frac{(-1)^r 2^r}{n^r}(2(\mathbf{x}^\top\mathbf{y})^2 + \|\mathbf{x}\|^2\|\mathbf{y}\|^2)\right.$$

$$\left. + \frac{(-1)^k 2^k}{n^{k-1}}\sum_{i=1}^{n}\mathbb{E}\left[(\mathbf{SD}_1\mathbf{x})_i^2(\mathbf{SD}_1\mathbf{y})_i^2\right]\right),$$

where we have used that $\mathbf{SD}_1$ is almost surely orthogonal, and therefore $\|\mathbf{SD}_1\mathbf{x}\|^2 = \|\mathbf{x}\|^2$ almost surely, $\|\mathbf{SD}_1\mathbf{y}\|^2 = \|\mathbf{y}\|^2$ almost surely, and $\langle\mathbf{SD}_1\mathbf{x}, \mathbf{SD}_1\mathbf{y}\rangle = \langle\mathbf{x}, \mathbf{y}\rangle$ almost surely. Applying Lemma 8.4 to the remaining expectation and collecting terms yields the required expression for $\operatorname{MSE}(\widehat{K}_m^{\mathcal{H},(k+1)}(\mathbf{x}, \mathbf{y}))$, and the proof is complete.

$\qquad\square$

## 8.6 Proof of Corollary 3.7

The proof follows simply by following the inductive strategy of the proof of Theorem 3.6, replacing the base case in Proposition 8.6 with the following.

**Proposition 8.7.** *Let $\mathbf{x}, \mathbf{y} \in \mathbb{R}^n$. The MSE of the single hybrid $\mathbf{SD}$-block $m$-feature estimator $K_m^{\mathcal{H},(1)}(\mathbf{x}, \mathbf{y})$ using a diagonal matrix with entries $\operatorname{Unif}(\{1, -1, i, -i\})$, rather than $\operatorname{Unif}(S^1)$ for $\langle\mathbf{x}, \mathbf{y}\rangle$ is*

$$\operatorname{MSE}(\widehat{K}_m^{\mathcal{H},(1)}(\mathbf{x}, \mathbf{y})) = \frac{1}{2m}\left(\langle\overline{\mathbf{x}}, \mathbf{x}\rangle\langle\overline{\mathbf{y}}, \mathbf{y}\rangle + \langle\overline{\mathbf{x}}, \mathbf{y}\rangle^2 - 2\sum_{r=1}^{n}x_r^2 y_r^2\right).$$

*Proof.* The proof of this proposition proceeds exactly as for Proposition 8.6; by following the same chain of reasoning, conditioning on the index set $\mathbf{J}$ of the sub-sampled rows, we arrive at

$$\mathrm{Var}\left(\mathrm{Re}\left(\sum_{p=1}^{m}(\mathbf{S}\overline{\mathbf{D}}_1\overline{\mathbf{x}})_{J_p}(\mathbf{S}\mathbf{D}_1\mathbf{y})_{J_p}\right)\,\Big|\,\mathbf{J}\right)=$$

$$\sum_{p,p'=1}^{m}\sum_{i,j,k,l=1}^{n}s_{J_p i}s_{J_p j}s_{J_{p'}k}s_{J_{p'}l}\mathrm{Cov}\left(\mathrm{Re}(\overline{d}_i\overline{x}_i d_j y_j),\mathrm{Re}(\overline{d}_k\overline{x}_k d_l y_l)\right).$$

Since we are dealing strictly with the case $\mathbf{x},\mathbf{y}\in\mathbb{R}^n$, we may simplify this further to obtain

$$\mathrm{Var}\left(\mathrm{Re}\left(\sum_{p=1}^{m}(\mathbf{S}\overline{\mathbf{D}}_1\overline{\mathbf{x}})_{J_p}(\mathbf{S}\mathbf{D}_1\mathbf{y})_{J_p}\right)\,\Big|\,\mathbf{J}\right)=$$

$$\sum_{p,p'=1}^{m}\sum_{i,j,k,l=1}^{n}s_{J_p i}s_{J_p j}s_{J_{p'}k}s_{J_{p'}l}x_i x_k y_i y_l\mathrm{Cov}\left(\mathrm{Re}(\overline{d}_i d_j),\mathrm{Re}(\overline{d}_k d_l)\right).$$

By calculating directly with the $d_i,d_j,d_k,d_l\sim\mathrm{Unif}(\{1,-1,i,-i\})$, we obtain

$$\mathrm{Var}\left(\mathrm{Re}\left(\sum_{p=1}^{m}(\mathbf{S}\overline{\mathbf{D}}_1\overline{\mathbf{x}})_{J_p}(\mathbf{S}\mathbf{D}_1\mathbf{y})_{J_p}\right)\,\Big|\,\mathbf{J}\right)=$$

$$\frac{1}{2}\sum_{p,p'=1}^{m}\sum_{i\neq j}^{n}s_{J_p i}s_{J_p j}s_{J_{p'}k}s_{J_{p'}l}(x_i^2 y_j^2 + x_i x_j y_i y_j),$$

exactly as in Proposition 8.6; following the rest of the argument of Proposition 8.6 yields the result. □

The proof of the corollary now follows by applying the steps of the proof of Theorem 3.6.

## 8.7 Exploring Dimensionality Reduction with Fully-complex Random Matrices

In this section, we briefly explore the possibility of using $\mathbf{SD}$-product matrices in which all the random diagonal matrices are complex-valued. Following on from the ROMs introduced in Definition 2.1, we define the $\mathbf{S}$-*Uniform* random matrix with $k\in\mathbb{N}$ blocks to be given by

$$\mathbf{M}_{\mathbf{S}\mathcal{U}}^{(k)}=\prod_{i=1}^{k}\mathbf{SD}_i^{(\mathcal{U})},$$

where $(\mathbf{D}_i^{(\mathcal{U})})_{i=1}^{k}$ are iid diagonal matrices with iid $\mathrm{Unif}(S^1)$ random variables on the diagonals, and $S^1$ is the unit circle of $\mathbb{C}$.

As alluded to in §3, we will see that introducing this increased number of complex parameters does not lead to significant increases in statistical performance relative to the estimator $\widehat{K}_m^{\mathcal{H},(k)}$ for dimensionality reduction.

We consider the estimator $\widehat{K}_m^{\mathcal{U},(k)}$ below, based on the sub-sampled $\mathbf{SD}$-product matrix $\mathbf{M}_{\mathbf{S}\mathcal{U}}^{(k),\mathrm{sub}}$:

$$\widehat{K}_m^{\mathcal{U},(k)}(\mathbf{x},\mathbf{y})=\frac{1}{m}\mathrm{Re}\left[\left(\overline{\mathbf{M}_{\mathbf{S}\mathcal{U}}^{(k),\mathrm{sub}}\mathbf{x}}\right)^{\top}\left(\mathbf{M}_{\mathbf{S}\mathcal{U}}^{(k),\mathrm{sub}}\mathbf{y}\right)\right],$$

and show that it does not yield a significant improvement over the estimator $\widehat{K}_m^{\mathcal{H},(k)}$ of Theorem 3.6:

**Theorem 8.8.** *For* $\mathbf{x},\mathbf{y}\in\mathbb{R}^n$, *the estimator* $\widehat{K}_m^{\mathcal{U},(k)}(\mathbf{x},\mathbf{y})$, *applying random sub-sampling strategy without replacement is unbiased and satisfies:*

$$\mathrm{MSE}(\widehat{K}_m^{\mathcal{U},(k)}(\mathbf{x},\mathbf{y}))=$$

$$\frac{1}{2m}\left(\frac{n-m}{n-1}\right)\left(\left((\mathbf{x}^{\top}\mathbf{y})^2+\|\mathbf{x}\|^2\|\mathbf{y}\|^2\right)+\sum_{r=1}^{k-1}\frac{(-1)^r}{n^r}(3(\mathbf{x}^{\top}\mathbf{y})^2+\|\mathbf{x}\|^2\|\mathbf{y}\|^2)+\frac{(-1)^k 2}{n^{k-1}}\sum_{i=1}^{n}x_i^2 y_i^2\right).$$

The structure of the proof of Theorem 8.8 is broadly the same as that of Theorem 3.3. We begin by remarking that the proof that the estimator is unbiased is exactly the same as that of Proposition 8.5. We then note that in the case of $k = 1$ block, the estimators $\widehat{K}_m^{\mathcal{H},(1)}$ and $\widehat{K}_m^{\mathcal{U},(1)}$, coincide so Proposition 8.6 establishes the MSE of the estimator $\widehat{K}_m^{\mathcal{U},(k)}$ in the base case $k = 1$. We then obtain a recursion formula for the MSE (Proposition 8.9), and finally prove the theorem by induction.

**Proposition 8.9.** *Let $k \geq 2$, $n \in \mathbb{N}$, $m \leq n$, and $\mathbf{x}, \mathbf{y} \in \mathbb{C}^n$ such that $\langle \overline{\mathbf{x}}, \mathbf{y} \rangle \in \mathbb{R}$; in particular, this includes $\mathbf{x}, \mathbf{y} \in \mathbb{R}^n$. Then we have the following recursion for the MSE of $\widehat{K}_M^{\mathcal{U},(k)}(\mathbf{x}, \mathbf{y})$:*

$$\mathrm{MSE}(\widehat{K}_m^{\mathcal{U},(k)}(\mathbf{x}, \mathbf{y})) = \mathbb{E}\left[ \mathrm{MSE}(\widehat{K}_m^{\mathcal{U},(k-1)}(\mathbf{SD}_1\mathbf{x}, \mathbf{SD}_1\mathbf{y})\big| \mathbf{D}_1) \right]$$

*Proof.* The proof is exactly analogous to that of Proposition 8.3, and is therefore omitted. □

Before we complete the proof by induction, we will need the following auxiliary result, to deal with the expectations that arise during the recursion due to the terms in the MSE expression of Proposition 8.6.

**Lemma 8.10.** *Under the assumptions of Theorem 8.8, we have the following expectations:*

$$\mathbb{E}\left[ |(\mathbf{SDx})_r|^2 |(\mathbf{SDy})_r|^2 \right] = \frac{1}{n^2} \left( \langle \overline{\mathbf{x}}, \mathbf{x} \rangle \langle \overline{\mathbf{y}}, \mathbf{y} \rangle + \langle \overline{\mathbf{x}}, \mathbf{y} \rangle^2 - \sum_{i=1}^{n} |x_i|^2 |y_i|^2 \right)$$

$$\mathbb{E}\left[ \mathrm{Re}((\mathbf{S\overline{D}x})_r^2 (\mathbf{SDy})_r^2) \right] = \frac{1}{n^2} \left( 2\langle \overline{\mathbf{x}}, \mathbf{y} \rangle^2 - \sum_{i=1}^{n} \mathrm{Re}(\overline{x}_i^2 y_i^2) \right)$$

*Proof.* For the first claim, we note that

$$\mathbb{E}\left[ |(\mathbf{SDx})_r|^2 |(\mathbf{SDy})_r|^2 \right] = \sum_{i,j,k,l}^{n} s_{ri} s_{rj} s_{rk} s_{rl} \overline{x}_i x_j \overline{y}_k y_l \mathbb{E}\left[ \overline{d}_i d_j \overline{d}_k d_l \right]$$

$$= \frac{1}{n^2} \left( \sum_{i \neq j} \overline{x}_i x_i \overline{y}_j y_j + \sum_{i \neq j} \overline{x}_i x_j \overline{y}_j y_i + \sum_{i=1}^{n} \overline{x}_i x_i \overline{y}_i y_i \right)$$

$$= \frac{1}{n^2} \left( \sum_{i,j=1}^{n} \overline{x}_i x_i \overline{y}_j y_j + \sum_{i,j=1}^{n} \overline{x}_i x_j \overline{y}_j y_i - \sum_{i=1}^{n} \overline{x}_i x_i \overline{y}_i y_i \right)$$

$$= \frac{1}{n^2} \left( \langle \overline{\mathbf{x}}, \mathbf{x} \rangle \langle \overline{\mathbf{y}}, \mathbf{y} \rangle + \langle \overline{\mathbf{x}}, \mathbf{y} \rangle^2 - \sum_{i=1}^{n} |x_i|^2 |y_i|^2 \right),$$

as required, where in the final equality we have use the assumption that $\langle \overline{\mathbf{x}}, \mathbf{y} \rangle \in \mathbb{R}$. For the second claim, we observe that

$$\mathbb{E}\left[ \mathrm{Re}((\mathbf{\overline{SD}x})_r^2 (\mathbf{SDy})_r^2) \right] = \mathrm{Re}\left( \sum_{i,j,k,l}^{n} s_{ri} s_{rj} s_{rk} s_{rl} \overline{x}_i \overline{x}_j y_k y_l \mathbb{E}\left[ \overline{d}_i \overline{d}_j d_k d_l \right] \right)$$

$$= \mathrm{Re}\left( \frac{1}{n^2} \left( 2 \sum_{i \neq j} \overline{x}_i \overline{x}_j y_i y_j + \sum_{i=1}^{n} \overline{x}_i \overline{x}_i y_i y_i \right) \right)$$

$$= \frac{1}{n^2} \left( 2\langle \overline{\mathbf{x}}, \mathbf{y} \rangle^2 - \sum_{i=1}^{n} \mathrm{Re}\left( \overline{x}_i^2 y_i^2 \right) \right),$$

where again we have used the assumption that $\langle \overline{\mathbf{x}}, \mathbf{y} \rangle \in \mathbb{R}$. □

*Proof of Theorem 8.8.* The proof now proceeds by induction. We in fact prove the stronger result that for any $\mathbf{x}, \mathbf{y} \in \mathbb{C}^n$ for which $\langle \overline{\mathbf{x}}, \mathbf{y} \rangle \in \mathbb{R}$, we have

$$\mathrm{MSE}(\widehat{K}_m^{\mathcal{U},(k)}(\mathbf{x}, \mathbf{y})) = \frac{1}{2m} \left( \frac{n-m}{n-1} \right) \left( (\langle \overline{\mathbf{x}}, \mathbf{y} \rangle^2 + \langle \overline{\mathbf{x}}, \mathbf{x} \rangle \langle \overline{\mathbf{y}}, \mathbf{y} \rangle) + \sum_{r=1}^{k-1} \frac{(-1)^r}{n^r} (3\langle \overline{\mathbf{x}}, \mathbf{y} \rangle^2 + \langle \overline{\mathbf{x}}, \mathbf{x} \rangle \langle \overline{\mathbf{y}}, \mathbf{y} \rangle) + $$

$$\frac{(-1)^k}{n^{k-1}}\left(\sum_{i=1}^n \left(|x_i|^2|y_i|^2 + \mathrm{Re}\left(\overline{x}_i^2 y_i^2\right)\right)\right).$$

from which Theorem 8.8 clearly follows. Proposition 8.6 yields the base case $k = 1$ for this claim. For the recursive step, suppose that the result holds for some number $k \in \mathbb{N}$ of blocks. Recalling the recursion of Proposition 8.9, we then obtain

$$\mathrm{MSE}(\widehat{K}_m^{\mathcal{U},(k+1)}(\mathbf{x},\mathbf{y})) = \frac{1}{2m}\left(\frac{n-m}{n-1}\right)\left(\left(\langle\overline{\mathbf{x}},\mathbf{y}\rangle^2 + \langle\overline{\mathbf{x}},\mathbf{x}\rangle\langle\overline{\mathbf{y}},\mathbf{y}\rangle\right) + \sum_{r=1}^{k-1}\frac{(-1)^r}{n^r}\left(3\langle\overline{\mathbf{x}},\mathbf{y}\rangle^2 + \langle\overline{\mathbf{x}},\mathbf{x}\rangle\langle\overline{\mathbf{y}},\mathbf{y}\rangle\right) + $$

$$\frac{(-1)^k}{n^{k-1}}\left(\sum_{i=1}^n \left(\mathbb{E}\left[|\mathbf{SD}_1\mathbf{x}|_i^2|\mathbf{SD}_1\mathbf{y}|_i^2\right] + \mathbb{E}\left[\mathrm{Re}\left((\overline{\mathbf{SD}_1\mathbf{x}})_i^2(\mathbf{SD}_1\mathbf{y})_i^2\right)\right]\right)\right)\right),$$

where we have used the fact that $\mathbf{SD}_1$ is a unitary isometry almost surely, and thus preserves Hermitian products. Applying Lemma 8.10 to the remaining expectations and collecting terms proves the inductive step, which concludes the proof of the theorem. $\qquad\square$

### 8.8 Proof of Theorem 3.8

*Proof.* The proof of this result is reasonably straightforward with the proofs of Theorems 3.3 and 3.6 in hand; we simply recognize where in these proofs the assumption of the sampling strategy without replacement was used. We deal first with Theorem 3.3, which deals with the MSE associated with $\widehat{K}_m^{(k)}(\mathbf{x},\mathbf{y})$. The only place in which the assumption of the sub-sampling strategy without replacement is used is mid-way through the proof of Proposition 8.2, which quantifies $\mathrm{MSE}(\widehat{K}_m^{(1)}(\mathbf{x},\mathbf{y}))$. Picking up the proof at the point the sub-sampling strategy is used, we have

$$\mathrm{MSE}(\widehat{K}_m^{(1)}(\mathbf{x},\mathbf{y})) = \frac{n^2}{m^2}\sum_{p,p'=1}^m\sum_{i\neq j}^n \left(x_i^2 y_j^2 + x_i x_j y_i y_j\right)\mathbb{E}\left[s_{J_p i}s_{J_p j}s_{J_{p'}i}s_{J_{p'}j}\right].$$

Now instead using sub-sampling strategy with replacement, note that each pair of sub-sampled indices $J_p$ and $J_{p'}$ are independent. Recalling that the columns of $\mathbf{S}$ are orthogonal, we obtain for distinct $p$ and $p'$ that

$$\mathbb{E}\left[s_{J_p i}s_{J_p j}s_{J_{p'}i}s_{J_{p'}j}\right] = \mathbb{E}\left[s_{J_p i}s_{J_p j}\right]\mathbb{E}\left[s_{J_{p'}i}s_{J_{p'}j}\right] = 0.$$

Again, for $p = p'$, we have $\mathbb{E}\left[s_{J_p i}s_{J_p j}s_{J_{p'}i}s_{J_{p'}j}\right] = 1/n^2$. Substituting the values of these expectations back into the expression for the MSE of $\widehat{K}_m^{(k)}(\mathbf{x},\mathbf{y})$ then yields

$$\mathrm{MSE}(\widehat{K}_m^{(1)}(\mathbf{x},\mathbf{y})) = \frac{n^2}{m^2}\sum_{i\neq j}^n \left(x_i^2 y_j^2 + x_i x_j y_i y_j\right)\left(m\times\frac{1}{n^2}\right)$$

$$= \frac{1}{m}\left(1 - \frac{m-1}{n-1}\right)\sum_{i\neq j}^n \left(x_i^2 y_j^2 + x_i x_j y_i y_j\right)$$

$$= \frac{1}{m}\left(\langle\mathbf{x},\mathbf{y}\rangle^2 + \|\mathbf{x}\|^2\|\mathbf{y}\|^2 - 2\sum_{i=1}^n x_i^2 y_i^2\right)$$

as required.

For the estimator $\widehat{K}_m^{\mathcal{H},(k)}(\mathbf{x},\mathbf{y})$, the result also immediately follows with the above calculation, as the only point in the proof of the MSE expressions for these estimators that is influenced by the sub-sampling strategy is in the calculation of the quantities $\mathbb{E}\left[s_{J_p i}s_{J_p j}s_{J_{p'}i}s_{J_{p'}j}\right]$; therefore, exactly the same multiplicative factor is incurred for MSE as for $\widehat{K}_m^{(k)}(\mathbf{x},\mathbf{y})$.

$\qquad\square$

## 9 Proofs of results in §4

### 9.1 Proof of Lemma 4.2

*Proof.* Follows immediately from the proof of Theorem 4.4 (see: the proof below). $\qquad\square$

## 9.2 Proof of Theorem 4.3

Recall that the angular kernel estimator based on $\mathbf{G}_{\text{ort}}$ is given by

$$\widehat{K}_m^{\text{ang,ort}}(\mathbf{x}, \mathbf{y}) = \frac{1}{m}\text{sign}(\mathbf{G}_{\text{ort}}\mathbf{x})^{\top}\text{sign}(\mathbf{G}_{\text{ort}}\mathbf{y})$$

where the function sign acts on vectors element-wise. In what follows, we write $\mathbf{G}_{\text{ort}}^i$ for the $i$th row of $\mathbf{G}_{\text{ort}}$, and $\mathbf{G}_i$ for the $i$th row of $\mathbf{G}$.

Since each $\mathbf{G}_{\text{ort}}^i$ has the same marginal distribution as $R_m$ in the unstructured Gaussian case covered by Theorem 4.4, unbiasedness of $\widehat{K}^{\text{ang,ort}}(x, y)$ follows immediately from this result, and so we obtain:

**Lemma 9.1.** $\widehat{K}_m^{\text{ang,ort}}(\mathbf{x}, \mathbf{y})$ *is an unbiased estimator of* $K^{\text{ang}}(\mathbf{x}, \mathbf{y})$.

We now turn our attention to the variance of $\widehat{K}_m^{\text{ang,ort}}(\mathbf{x}, \mathbf{y})$.

**Theorem 9.2.** *The variance of the estimator* $\widehat{K}_m^{\text{ang,ort}}(x, y)$ *is strictly smaller than the variance of* $\widehat{K}_m^{\text{ang, base}}(\mathbf{x}, \mathbf{y})$

*Proof.* Denote by $\theta$ the angle between $\mathbf{x}$ and $\mathbf{y}$, and for notational ease, let $S_i = \text{sign}\left(\langle \mathbf{G}^i, \mathbf{x}\rangle\right)\text{sign}\left(\langle \mathbf{G}^i, \mathbf{y}\rangle\right)$, and $S_i^{\text{ort}} = \text{sign}\left(\langle \mathbf{G}_{\text{ort}}^i, \mathbf{x}\rangle\right)\text{sign}\left(\langle \mathbf{G}_{\text{ort}}^i, \mathbf{y}\rangle\right)$. Now observe that as $\widehat{K}_m^{\text{ang,ort}}(\mathbf{x}, \mathbf{y})$ is unbiased, we have

$$\text{Var}\left(\widehat{K}_m^{\text{ang,ort}}(\mathbf{x}, \mathbf{y})\right)$$
$$= \text{Var}\left(\frac{1}{m}\sum_{i=1}^{m} S_i^{\text{ort}}\right)$$
$$= \frac{1}{m^2}\left(\sum_{i=1}^{m}\text{Var}\left(S_i^{\text{ort}}\right) + \sum_{i\neq i'}^{m}\text{Cov}\left(S_i^{\text{ort}}, S_{i'}^{\text{ort}}\right)\right).$$

By a similar argument, we have

$$\text{Var}\left(\widehat{K}_m^{\text{base}}(\mathbf{x}, \mathbf{y})\right) = \frac{1}{m^2}\left(\sum_{i=1}^{m}\text{Var}\left(S_i\right) + \sum_{i\neq i'}^{m}\text{Cov}\left(S_i, S_{i'}\right)\right). \tag{29}$$

Note that the covariance terms in (29) evaluate to 0, by independence of $S_i$ and $S_{i'}$ for $i \neq i'$ (which is inherited from the independence of $\mathbf{G}^i$ and $\mathbf{G}^{i'}$). Also observe that since $\mathbf{G}^i \overset{d}{=} \mathbf{G}_{\text{ort}}^i$, we have

$$\text{Var}\left(S_i^{\text{ort}}\right) = \text{Var}\left(S_i\right).$$

Therefore, demonstrating the theorem is equivalent to showing, for $i \neq i'$, that

$$\text{Cov}\left(S_i^{\text{ort}}, S_{i'}^{\text{ort}}\right) < 0,$$

which is itself equivalent to showing

$$\mathbb{E}\left[S_i^{\text{ort}} S_{i'}^{\text{ort}}\right] < \mathbb{E}\left[S_i^{\text{ort}}\right]\mathbb{E}\left[S_{i'}^{\text{ort}}\right]. \tag{30}$$

Note that the variables $(S_i^{\text{ort}})_{i=1}^m$ take values in $\{\pm 1\}$. Denoting $\mathcal{A}_i = \{S_i^{\text{ort}} = -1\}$ for $i = 1, \ldots, m$, we can rewrite (30) as

$$\mathbb{P}\left[\mathcal{A}_i^c \cap \mathcal{A}_{i'}^c\right] + \mathbb{P}\left[\mathcal{A}_i \cap \mathcal{A}_{i'}\right] - \mathbb{P}\left[\mathcal{A}_i \cap \mathcal{A}_{i'}^c\right] - \mathbb{P}\left[\mathcal{A}_i^c \cap \mathcal{A}_{i'}\right] < \left(\frac{\pi - 2\theta}{\pi}\right)^2.$$

Note that the left-hand side is equal to

$$2(\mathbb{P}\left[\mathcal{A}_i^c \cap \mathcal{A}_{i'}^c\right] + \mathbb{P}\left[\mathcal{A}_i \cap \mathcal{A}_{i'}\right]) - 1.$$

Plugging in the bounds of Proposition 9.3, and using the fact that the pair of indicators $(\mathbb{1}_{\mathcal{A}_i}, \mathbb{1}_{\mathcal{A}_{i'}})$ is identically distributed for all pairs of distinct indices $i, i' \in \{1, \ldots, m\}$, thus yields the result. $\quad\square$

**Proposition 9.3.** *We then have the following inequalities:*

$$\mathbb{P}\left[\mathcal{A}_1 \cap \mathcal{A}_2\right] < \left(\frac{\theta}{\pi}\right)^2 \qquad \text{and} \qquad \mathbb{P}\left[\mathcal{A}_1^c \cap \mathcal{A}_2^c\right] < \left(1 - \frac{\theta}{\pi}\right)^2$$

Before providing the proof of this proposition, we describe some coordinate choices we will make in order to obtain the bounds in Proposition 9.3.

We pick an orthonormal basis for $\mathbb{R}^n$ so that the first two coordinates span the **x-y** plane, and further so that $(\mathbf{G}_{\text{ort}}^1)_2$, the coordinate of $\mathbf{G}_{\text{ort}}^1$ in the second dimension, is 0. We extend this to an orthonormal basis of $\mathbb{R}^n$ so that $(\mathbf{G}_{\text{ort}}^1)_3 \geq 0$, and $(\mathbf{G}_{\text{ort}}^1)_i = 0$ for $i \geq 4$. Thus, in this basis, we have coordinates

$$\mathbf{G}_{\text{ort}}^1 = ((\mathbf{G}_{\text{ort}}^1)_1, 0, (\mathbf{G}_{\text{ort}}^1)_3, 0, \dots, 0),$$

with $(\mathbf{G}_{\text{ort}}^1)_1 \sim \chi_2$ and $(\mathbf{G}_{\text{ort}}^1)_3 \sim \chi_{N-2}$ (by elementary calculations with multivariate Gaussian distributions). Note that the angle, $\phi$, that $\mathbf{G}_{\text{ort}}^1$ makes with the **x-y** plane is then $\phi = \arctan((\mathbf{G}_{\text{ort}}^1)_3/(\mathbf{G}_{\text{ort}}^1)_1)$. Having fixed our coordinate system relative to the random variable $\mathbf{G}_{\text{ort}}^1$, the coordinates of **x** and **y** in this frame are now themselves random variables; we introduce the angle $\psi$ to describe the angle between **x** and the positive first coordinate axis in this basis.

Now consider $\mathbf{G}_{\text{ort}}^2$. We are concerned with the direction of $((\mathbf{G}_{\text{ort}}^2)_1, (\mathbf{G}_{\text{ort}}^2)_2)$ in the **x-y** plane. Conditional on $\mathbf{G}_{\text{ort}}^1$, the direction of the full vector $\mathbf{G}_{\text{ort}}^2$ is distributed uniformly on $S^{n-2}(\langle \mathbf{G}_{\text{ort}}^1 \rangle^\perp)$, the set of unit vectors orthogonal to $\mathbf{G}_{\text{ort}}^1$. Because of our particular choice of coordinates, we can therefore write

$$\mathbf{G}_{\text{ort}}^2 = (r\sin(\phi), (\mathbf{G}_{\text{ort}}^2)_2, r\cos(\phi), (\mathbf{G}_{\text{ort}}^2)_4, (\mathbf{G}_{\text{ort}}^2)_5, \dots, (\mathbf{G}_{\text{ort}}^2)_n),$$

where the $(N-1)$-dimensional vector $(r, (\mathbf{G}_{\text{ort}}^2)_2, (\mathbf{G}_{\text{ort}}^2)_4, (\mathbf{G}_{\text{ort}}^2)_5, \dots, (\mathbf{G}_{\text{ort}}^2)_n)$ has an isotropic distribution.

So the direction of $((\mathbf{G}_{\text{ort}}^2)_1, (\mathbf{G}_{\text{ort}}^2)_2)$ in the **x-y** plane follows an angular Gaussian distribution, with covariance matrix

$$\begin{pmatrix} \sin^2(\phi) & 0 \\ 0 & 1 \end{pmatrix}.$$

With these geometrical considerations in place, we are ready to give the proof of Proposition 9.3.

*Proof of Proposition 9.3.* Dealing with the first inequality, we decompose the event as

$$\begin{aligned} \mathcal{A}_1 \cap \mathcal{A}_2 =& \{\langle \mathbf{G}_{\text{ort}}^1, \mathbf{x} \rangle > 0, \langle \mathbf{G}_{\text{ort}}^1, \mathbf{y} \rangle < 0, \langle \mathbf{G}_{\text{ort}}^2, \mathbf{x} \rangle > 0, \langle \mathbf{G}_{\text{ort}}^2, \mathbf{y} \rangle < 0\} \\ &\cup \{\langle \mathbf{G}_{\text{ort}}^1, \mathbf{x} \rangle > 0, \langle \mathbf{G}_{\text{ort}}^1, \mathbf{y} \rangle < 0, \langle \mathbf{G}_{\text{ort}}^2, \mathbf{x} \rangle < 0, \langle \mathbf{G}_{\text{ort}}^2, \mathbf{y} \rangle > 0\} \\ &\cup \{\langle \mathbf{G}_{\text{ort}}^1, \mathbf{x} \rangle < 0, \langle \mathbf{G}_{\text{ort}}^1, \mathbf{y} \rangle > 0, \langle \mathbf{G}_{\text{ort}}^2, \mathbf{x} \rangle > 0, \langle \mathbf{G}_{\text{ort}}^2, \mathbf{y} \rangle < 0\} \\ &\cup \{\langle \mathbf{G}_{\text{ort}}^1, \mathbf{x} \rangle < 0, \langle \mathbf{G}_{\text{ort}}^1, \mathbf{y} \rangle > 0, \langle \mathbf{G}_{\text{ort}}^2, \mathbf{x} \rangle < 0, \langle \mathbf{G}_{\text{ort}}^2, \mathbf{y} \rangle > 0\}. \end{aligned}$$

As the law of $(\mathbf{G}_{\text{ort}}^1, \mathbf{G}_{\text{ort}}^2)$ is the same as that of $(\mathbf{G}_{\text{ort}}^2, \mathbf{G}_{\text{ort}}^1)$ and that of $(-\mathbf{G}_{\text{ort}}^1, \mathbf{G}_{\text{ort}}^2)$, it follows that all four events in the above expression have the same probability. The statement of the theorem is therefore equivalent to demonstrating the following inequality:

$$\mathbb{P}\left[\langle \mathbf{G}_{\text{ort}}^1, x \rangle > 0, \langle \mathbf{G}_{\text{ort}}^1, \mathbf{y} \rangle < 0, \langle \mathbf{G}_{\text{ort}}^2, \mathbf{x} \rangle > 0, \langle \mathbf{G}_{\text{ort}}^2, \mathbf{y} \rangle < 0\right] < \left(\frac{\theta}{2\pi}\right)^2.$$

We now proceed according to the coordinate choices described above. We first condition on the random angles $\phi$ and $\psi$ to obtain

$$\begin{aligned} &\mathbb{P}\left[\langle \mathbf{G}_{\text{ort}}^1, \mathbf{x} \rangle > 0, \langle \mathbf{G}_{\text{ort}}^1, \mathbf{y} \rangle < 0, \langle \mathbf{G}_{\text{ort}}^2, \mathbf{x} \rangle > 0, \langle \mathbf{G}_{\text{ort}}^2, \mathbf{y} \rangle < 0\right] \\ &= \int_0^{2\pi} \frac{\mathrm{d}\psi}{2\pi} \int_0^{\pi/2} f(\phi)\mathrm{d}\phi \, \mathbb{P}\left[\langle \mathbf{G}_{\text{ort}}^1, \mathbf{x} \rangle > 0, \langle \mathbf{G}_{\text{ort}}^1, \mathbf{y} \rangle < 0, \langle \mathbf{G}_{\text{ort}}^2, \mathbf{x} \rangle > 0, \langle \mathbf{G}_{\text{ort}}^2, \mathbf{y} \rangle < 0 | \psi, \phi\right] \\ &= \int_0^{2\pi} \frac{\mathrm{d}\psi}{2\pi} \int_0^{\pi/2} f(\phi)\mathrm{d}\phi \, \mathbb{1}_{\{0 \in [\psi - \pi/2, \psi - \pi/2 + \theta]\}} \mathbb{P}\left[\langle \mathbf{G}_{\text{ort}}^2, \mathbf{x} \rangle > 0, \langle \mathbf{G}_{\text{ort}}^2, \mathbf{y} \rangle < 0 | \psi, \phi\right], \end{aligned}$$

where $f$ is the density of the random angle $\phi$. The final equality above follows as $\mathbf{G}_{\mathrm{ort}}^1$ and $\mathbf{G}_{\mathrm{ort}}^2$ are independent conditional on $\psi$ and $\phi$, and since the event $\{\langle \mathbf{G}_{\mathrm{ort}}^1, \mathbf{x} \rangle > 0, \langle \mathbf{G}_{\mathrm{ort}}^1, \mathbf{y} \rangle < 0\}$ is exactly the event $\{0 \in [\psi - \pi/2, \psi - \pi/2 + \theta]\}$, by considering the geometry of the situation in the $\mathbf{x}$-$\mathbf{y}$ plane. We can remove the indicator function from the integrand by adjusting the limits of integration, obtaining

$$\mathbb{P}\left[\langle \mathbf{G}_{\mathrm{ort}}^1, \mathbf{x} \rangle > 0, \langle \mathbf{G}_{\mathrm{ort}}^1, \mathbf{y} \rangle < 0, \langle \mathbf{G}_{\mathrm{ort}}^2, \mathbf{x} \rangle > 0, \langle \mathbf{G}_{\mathrm{ort}}^2, \mathbf{y} \rangle < 0\right]$$
$$= \int_{\pi/2-\theta}^{\pi/2} \frac{\mathrm{d}\psi}{2\pi} \int_0^{\pi/2} f(\phi)\mathrm{d}\phi \, \mathbb{P}\left[\langle \mathbf{G}_{\mathrm{ort}}^2, \mathbf{x} \rangle > 0, \langle \mathbf{G}_{\mathrm{ort}}^2, \mathbf{y} \rangle < 0 | \psi, \phi\right] .$$

We now turn our attention to the conditional probability

$$\mathbb{P}\left[\langle \mathbf{G}_{\mathrm{ort}}^2, \mathbf{x} \rangle > 0, \langle \mathbf{G}_{\mathrm{ort}}^2, \mathbf{y} \rangle < 0 | \psi, \phi\right] .$$

The event $\{\langle \mathbf{G}_{\mathrm{ort}}^2, \mathbf{x} \rangle > 0, \langle \mathbf{G}_{\mathrm{ort}}^2, \mathbf{y} \rangle < 0\}$ is equivalent to the angle $t$ of the projection of $\mathbf{G}_{\mathrm{ort}}^2$ into the $\mathbf{x}$-$\mathbf{y}$ plane with the first coordinate axis lying in the interval $[\psi - \pi/2, \psi - \pi/2 + \theta]$. Recalling the distribution of the angle $t$ from the geometric considerations described immediately before this proof, we obtain

$$\mathbb{P}\left[\langle \mathbf{G}_{\mathrm{ort}}^1, \mathbf{x} \rangle > 0, \langle \mathbf{G}_{\mathrm{ort}}^1, \mathbf{y} \rangle < 0, \langle \mathbf{G}_{\mathrm{ort}}^2, \mathbf{x} \rangle > 0, \langle \mathbf{G}_{\mathrm{ort}}^2, \mathbf{y} \rangle < 0\right]$$
$$= \int_{\pi/2-\theta}^{\pi/2} \frac{\mathrm{d}\psi}{2\pi} \int_0^{\pi/2} f(\phi)\mathrm{d}\phi \int_{\psi-\pi/2}^{\psi-\pi/2+\theta} (2\pi \sin(\phi))^{-1}(\cos^2(t)/\sin^2(\phi) + \sin^2(t))^{-1}dt .$$

With $\theta \in [0, \pi/2]$, we note that the integral with respect to $t$ can be evaluated analytically, leading us to

$$\mathbb{P}\left[\langle \mathbf{G}_{\mathrm{ort}}^1, \mathbf{x} \rangle > 0, \langle \mathbf{G}_{\mathrm{ort}}^1, \mathbf{y} \rangle < 0, \langle \mathbf{G}_{\mathrm{ort}}^2, \mathbf{x} \rangle > 0, \langle \mathbf{G}_{\mathrm{ort}}^2, \mathbf{y} \rangle < 0\right]$$
$$= \int_{\pi/2-\theta}^{\pi/2} \frac{\mathrm{d}\psi}{2\pi} \int_0^{\pi/2} f(\phi)\mathrm{d}\phi \, \frac{1}{2\pi} \left(\arctan(\tan(\psi - \pi/2 + \theta)\sin(\phi)) - \arctan(\tan(\psi - \pi/2)\sin(\phi))\right)$$
$$\leq \int_{\pi/2-\theta}^{\pi/2} \frac{\mathrm{d}\psi}{2\pi} \int_0^{\pi/2} f(\phi)\mathrm{d}\phi \, \frac{\theta}{2\pi}$$
$$= \left(\frac{\theta}{2\pi}\right)^2 .$$

To deal with $\theta \in [\pi/2, \pi]$, we note that if the angle $\theta$ between $\mathbf{x}$ and $\mathbf{y}$ is obtuse, then the angle between $\mathbf{x}$ and $-\mathbf{y}$ is $\pi - \theta$ and therefore acute. Recalling from our definition that $\mathcal{A}_m = \{\mathrm{sign}\left(\langle \mathbf{G}_{\mathrm{ort}}^i, \mathbf{x} \rangle\right) \mathrm{sign}\left(\langle \mathbf{G}_{\mathrm{ort}}^i, \mathbf{y} \rangle\right) = -1\}$, if we denote the corresponding quantity for the pair of vecors $\mathbf{x}$, $-\mathbf{y}$ by $\bar{\mathcal{A}}_m = \{\mathrm{sign}\left(\langle \mathbf{G}_{\mathrm{ort}}^i, \mathbf{x} \rangle\right) \mathrm{sign}\left(\langle \mathbf{G}_{\mathrm{ort}}^i, -\mathbf{y} \rangle\right) = -1\}$, then we in fact have $\bar{\mathcal{A}}_m = \mathcal{A}_m^c$. Therefore, applying the result to the pair of vectors $\mathbf{x}$ and $-\mathbf{y}$ (which have acute angle $\pi - \theta$ between them) and using the inclusion-exclusion principle, we obtain:

$$\mathbb{P}(\mathcal{A}_1 \cap \mathcal{A}_2) = 1 - \mathbb{P}(\mathcal{A}_1^c) - \mathbb{P}(\mathcal{A}_2^c) + \mathbb{P}(\mathcal{A}_1^c \cap \mathcal{A}_2^c)$$
$$< 1 - \mathbb{P}(\mathcal{A}_1^c) - \mathbb{P}(\mathcal{A}_2^c) + \left(\frac{\pi - \theta}{\pi}\right)^2$$
$$= 1 - 2\left(\frac{\pi - \theta}{\pi}\right) + \left(\frac{\pi - \theta}{\pi}\right)^2$$
$$= \left(\frac{\theta}{\pi}\right)^2 .$$

as required.

The second inequality of Proposition 9.3 follows from the inclusion-exclusion principle and the first inequality:

$$\mathbb{P}\left[\mathcal{A}_1^c \cap \mathcal{A}_2^c\right] = 1 - \mathbb{P}\left[\mathcal{A}_1\right] - \mathbb{P}\left[\mathcal{A}_2\right] + \mathbb{P}\left[\mathcal{A}_1 \cap \mathcal{A}_2\right]$$
$$< 1 - \mathbb{P}\left[\mathcal{A}_1\right] - \mathbb{P}\left[\mathcal{A}_2\right] + \left(\frac{\theta}{\pi}\right)^2$$

$$= (1 - \mathbb{P}\left[\mathcal{A}_1\right])(1 - \mathbb{P}\left[\mathcal{A}_2\right])$$

$$= \left(1 - \frac{\theta}{\pi}\right)^2.$$

$\square$

## 9.3 Proof of Theorem 4.4

*Proof.* We will consider the following setting. Given two vectors $\mathbf{x}, \mathbf{y} \in \mathbb{R}^n$, each of them is transformed by the nonlinear mapping: $\phi^{\mathbf{M}} : \mathbf{z} \to \frac{1}{\sqrt{k}} \mathrm{sgn}(\mathbf{Mz})$, where $\mathbf{M} \in \mathbb{R}^{m \times n}$ is some linear transformation and $\mathrm{sgn}(\mathbf{v})$ stands for a vector obtained from $\mathbf{v}$ by applying pointwise nonlinear mapping $\mathrm{sgn} : \mathbb{R} \to \mathbb{R}$ defined as follows: $\mathrm{sgn}(x) = +1$ if $x > 0$ and $\mathrm{sgn}(x) = -1$ otherwise. The angular distance $\theta$ between $\mathbf{x}$ and $\mathbf{y}$ is estimated by: $\hat{\theta}^{\mathbf{M}} = \frac{\pi}{2}(1 - \phi^{\mathbf{M}}(\mathbf{x})^\top \phi^{\mathbf{M}}(\mathbf{y}))$. We will derive the formula for the $\mathrm{MSE}(\hat{\theta}^{\mathbf{M}}(\mathbf{x}, \mathbf{y}))$. One can easily see that the MSE of the considered in the statement of the theorem angular kernel on vectors $\mathbf{x}$ and $\mathbf{y}$ can be obtained from this one by multiplying by $\frac{4}{\pi^2}$.

Denote by $\mathbf{r}^i$ the $i^{th}$ row of $\mathbf{M}$. Notice first that for any two vectors $\mathbf{x}, \mathbf{y} \in \mathbb{R}^n$ with angular distance $\theta$, the event $E_i = \{\mathrm{sgn}((\mathbf{r}^i)^\top \mathbf{x}) \neq \mathrm{sgn}((\mathbf{r}^i)^\top \mathbf{y})\}$ is equivalent to the event $\{\mathbf{r}^i_{proj} \in \mathcal{R}\}$, where $\mathbf{r}^i_{proj}$ stands for the projection of $\mathbf{r}^i$ into the $\mathbf{x} - \mathbf{y}$ plane and $\mathcal{R}$ is a union of two cones in the $\mathbf{x}$-$\mathbf{y}$ plane obtained by rotating vectors $\mathbf{x}$ and $\mathbf{y}$ by $\frac{\pi}{2}$. Denote $\mathcal{A}^i = \{\mathbf{r}^i_{proj} \in \mathcal{R}\}$ for $i = 1, ..., k$ and $\delta_{i,j} = \mathbb{P}[\mathcal{A}^i \cap \mathcal{A}^j] - \mathbb{P}[\mathcal{A}^i]\mathbb{P}[\mathcal{A}^j]$.

For a warmup, let us start our analysis for the standard unstructured Gaussian estimator case. It is a well known fact that this is an unbiased estimator of $\theta$. Thus

$$\mathrm{MSE}(\hat{\theta}^{\mathbf{G}}(\mathbf{x}, \mathbf{y})) = Var(\frac{\pi}{2}(1 - \phi^{\mathbf{M}}(\mathbf{x})^\top \phi^{\mathbf{M}}(\mathbf{y}))) = \frac{\pi^2}{4} Var(\phi^{\mathbf{M}}(\mathbf{x})^\top \phi^{\mathbf{M}}(\mathbf{y}))) \\ = \frac{\pi^2}{4} \frac{1}{m^2} Var(\sum_{i=1}^m X_i), \tag{31}$$

where $X_i = \mathrm{sgn}((\mathbf{r}^i)^\top \mathbf{x})\mathrm{sgn}((\mathbf{r}^i)^\top \mathbf{y})$.

Since the rows of $\mathbf{G}$ are independent, we get

$$Var(\sum_{i=1}^m X_i) = \sum_{i=1}^m Var(X_i) = \sum_{i=1}^m (\mathbb{E}[X_i^2] - \mathbb{E}[X_i]^2). \tag{32}$$

From the unbiasedness of the estimator, we have: $\mathbb{E}[X_i] = (-1) \cdot \frac{\theta}{\pi} + 1 \cdot (1 - \frac{\theta}{\pi})$. Thus we get:

$$\mathrm{MSE}(\hat{\theta}^{\mathbf{G}}(\mathbf{x}, \mathbf{y})) = \frac{\pi^2}{4} \frac{1}{m^2} \sum_{i=1}^m (1 - (1 - \frac{2\theta}{\pi})^2) = \frac{\theta(\pi - \theta)}{m}. \tag{33}$$

Multiplying by $\frac{4}{\pi^2}$, we obtain the proof of Lemma 4.2.

Now let us switch to the general case. We first compute the variance of the general estimator $\mathcal{E}$ using matrices $\mathbf{M}$ (note that in this setting we do not assume that the estimator is necessarily unbiased).

By the same analysis as before, we get:

$$Var(\mathcal{E}) = Var(\frac{\pi}{2}(1 - \phi(\mathbf{x})^\top \phi(\mathbf{y}))) = \frac{\pi^2}{4} Var(\phi(\mathbf{x})^\top \phi(\mathbf{y}))) = \frac{\pi^2}{4} \frac{1}{m^2} Var(\sum_{i=1}^m X_i), \tag{34}$$

This time however different $X_i$s are not uncorrelated. We get

$$Var(\sum_{i=1}^{m} X_i) = \sum_{i=1}^{m} Var(X_i) + \sum_{i \neq j} Cov(X_i, X_j) =$$

$$\sum_{i=1}^{m} \mathbb{E}[X_i^2] - \sum_{i=1}^{m} \mathbb{E}[X_i]^2 + \sum_{i \neq j} \mathbb{E}[X_i X_j] - \sum_{i \neq j} \mathbb{E}[X_i]\mathbb{E}[X_j] = \tag{35}$$

$$m + \sum_{i \neq j} \mathbb{E}[X_i X_j] - \sum_{i,j} \mathbb{E}[X_i]\mathbb{E}[X_j]$$

Now, notice that from our previous observations and the definition of $\mathcal{A}^i$, we have

$$\mathbb{E}[X_i] = -\mathbb{P}[\mathcal{A}^i] + \mathbb{P}[\mathcal{A}_c^i], \tag{36}$$

where $\mathcal{A}_c^i$ stands for the complement of $\mathcal{A}^i$.

By the similar analysis, we also get:

$$\mathbb{E}[X_i X_j] = \mathbb{P}[\mathcal{A}^i \cap \mathcal{A}^j] + \mathbb{P}[\mathcal{A}_c^i \cap \mathcal{A}_c^j] - \mathbb{P}[\mathcal{A}_c^i \cap \mathcal{A}^j] - \mathbb{P}[\mathcal{A}^i \cap \mathcal{A}_c^j] \tag{37}$$

Thus we obtain

$$Var(\sum_{i=1}^{m} X_i) = m + \sum_{i \neq j}(\mathbb{P}[\mathcal{A}^i \cap \mathcal{A}^j] + \mathbb{P}[\mathcal{A}_c^i \cap \mathcal{A}_c^j] - \mathbb{P}[\mathcal{A}_c^i \cap \mathcal{A}^j] - \mathbb{P}[\mathcal{A}^i \cap \mathcal{A}_c^j]$$

$$-(\mathbb{P}[\mathcal{A}_c^i] - \mathbb{P}[\mathcal{A}^i])(\mathbb{P}[\mathcal{A}_c^j] - \mathbb{P}[\mathcal{A}^j]))$$

$$-\sum_{i}(\mathbb{P}[\mathcal{A}_c^i] - \mathbb{P}[\mathcal{A}^i])^2 = m - \sum_{i}(1 - 2\mathbb{P}[\mathcal{A}^i])^2$$

$$+\sum_{i \neq j}(\mathbb{P}[\mathcal{A}^i \cap \mathcal{A}^j] + \mathbb{P}[\mathcal{A}_c^i \cap \mathcal{A}_c^j] - \mathbb{P}[\mathcal{A}_c^i \cap \mathcal{A}^j] - \mathbb{P}[\mathcal{A}^i \cap \mathcal{A}_c^j] + \tag{38}$$

$$\mathbb{P}[\mathcal{A}_c^i]\mathbb{P}[\mathcal{A}^j] + \mathbb{P}[\mathcal{A}^i]\mathbb{P}[\mathcal{A}_c^j] - \mathbb{P}[\mathcal{A}_c^i]\mathbb{P}[\mathcal{A}_c^j] - \mathbb{P}[\mathcal{A}^i]\mathbb{P}[\mathcal{A}^j])$$

$$= m - \sum_{i}(1 - 2\mathbb{P}[\mathcal{A}^i])^2 + \sum_{i \neq j}(\delta_1(i,j) + \delta_2(i,j) + \delta_3(i,j) + \delta_4(i,j)),$$

where

- $\delta_1(i,j) = \mathbb{P}[\mathcal{A}^i \cap \mathcal{A}^j] - \mathbb{P}[\mathcal{A}^i]\mathbb{P}[\mathcal{A}^j]$,
- $\delta_2(i,j) = \mathbb{P}[\mathcal{A}_c^i \cap \mathcal{A}_c^j] - \mathbb{P}[\mathcal{A}_c^i]\mathbb{P}[\mathcal{A}_c^j]$,
- $\delta_3(i,j) = \mathbb{P}[\mathcal{A}_c^i]\mathbb{P}[\mathcal{A}^j] - \mathbb{P}[\mathcal{A}_c^i \cap \mathcal{A}^j]$,
- $\delta_4(i,j) = \mathbb{P}[\mathcal{A}^i]\mathbb{P}[\mathcal{A}_c^j] - \mathbb{P}[\mathcal{A}^i \cap \mathcal{A}_c^j]$.

Now note that

$$-\delta_4(i,j) = \mathbb{P}[\mathcal{A}^i] - \mathbb{P}[\mathcal{A}^i \cap \mathcal{A}^j] - \mathbb{P}[\mathcal{A}^i]\mathbb{P}[\mathcal{A}_c^j]$$

$$= \mathbb{P}[\mathcal{A}^i] - \mathbb{P}[\mathcal{A}^i](1 - \mathbb{P}[\mathcal{A}^j]) - \mathbb{P}[\mathcal{A}^i \cap \mathcal{A}^j] \tag{39}$$

$$= \mathbb{P}[\mathcal{A}^i]\mathbb{P}[\mathcal{A}^j] - \mathbb{P}[\mathcal{A}^i \cap \mathcal{A}^j] = -\delta_1(i,j)$$

Thus we have $\delta_4(i,j) = \delta_1(i,j)$. Similarly, $\delta_3(i,j) = \delta_1(i,j)$. Notice also that

$$-\delta_2(i,j) = (1 - \mathbb{P}[\mathcal{A}^i])(1 - \mathbb{P}[\mathcal{A}^j]) - (\mathbb{P}[\mathcal{A}_c^i] - \mathbb{P}[\mathcal{A}_c^i \cap \mathcal{A}^j])$$

$$= 1 - \mathbb{P}[\mathcal{A}^i] - \mathbb{P}[\mathcal{A}^j] + \mathbb{P}[\mathcal{A}^i]\mathbb{P}[\mathcal{A}^j] - 1 + \mathbb{P}[\mathcal{A}^i] + \mathbb{P}[\mathcal{A}_c^i \cap \mathcal{A}^j] \tag{40}$$

$$= \mathbb{P}[\mathcal{A}^i]\mathbb{P}[\mathcal{A}^j] - \mathbb{P}[\mathcal{A}^i \cap \mathcal{A}^j] = -\delta_1(i,j),$$

therefore $\delta_2(i,j) = \delta_1(i,j)$.

Thus, if we denote $\delta_{i,j} = \delta_1(i,j) = \mathbb{P}[\mathcal{A}^i \cap \mathcal{A}^j] - \mathbb{P}[\mathcal{A}^i]\mathbb{P}[\mathcal{A}^j]$, then we get

$$Var(\sum_{i=1}^{m} X_i) = m - \sum_{i}(1 - 2\mathbb{P}[A^i])^2 + 4\sum_{i \neq j} \delta_{i,j}. \tag{41}$$

Thus we obtain

$$Var(\mathcal{E}) = \frac{\pi^2}{4m^2}[m - \sum_i(1 - 2\mathbb{P}[A^i])^2 + 4\sum_{i \neq j}\delta_{i,j}]. \tag{42}$$

Note that $Var(\mathcal{E}) = \mathbb{E}[(\mathcal{E} - \mathbb{E}[\mathcal{E}])^2]$. We have:

$$\begin{aligned}
\text{MSE}(\hat{\theta}^{\mathbf{M}}(\mathbf{x}, \mathbf{y})) = \mathbb{E}[(\mathcal{E} - \theta)^2] &= \mathbb{E}[(\mathcal{E} - \mathbb{E}[\mathcal{E}])^2] + \mathbb{E}[(\mathcal{E} - \theta)^2] - \mathbb{E}[(\mathcal{E} - \mathbb{E}[\mathcal{E}])^2] \\
&= Var(\mathcal{E}) + \mathbb{E}[(\mathcal{E} - \theta)^2 - (\mathcal{E} - \mathbb{E}[\mathcal{E}])^2] \\
&= Var(\mathcal{E}) + (\mathbb{E}[\mathcal{E}] - \theta)^2
\end{aligned} \tag{43}$$

Notice that $\mathcal{E} = \frac{\pi}{2}(1 - \frac{1}{m}\sum_{i=1}^m X_i)$. Thus we get:

$$\text{MSE}(\hat{\theta}^{\mathbf{M}}(\mathbf{x}, \mathbf{y})) = \frac{\pi^2}{4m^2}[m - \sum_i(1 - 2\mathbb{P}[A^i])^2 + 4\sum_{i \neq j}\delta_{i,j}] + \frac{\pi^2}{m^2}\sum_i(\mathbb{P}(\mathcal{A}^i) - \frac{\theta}{\pi})^2. \tag{44}$$

Now it remains to multiply the expression above by $\frac{4}{\pi^2}$ and that completes the proof. $\square$

**Remark 9.4.** *Notice that if $\mathbb{P}(\mathcal{A}^i) = \frac{\theta}{\pi}$ (this is the case for the standard unstructured estimator as well as for the considered by us estimator using orthogonalized version of Gaussian vectors) and if rows of matrix $\mathbf{M}$ are independent then the general formula for* MSE *for the estimator of an angle reduces to $\frac{(\pi-\theta)\theta}{m}$. If the first property is satisfied but the rows are not necessarily independent (as it is the case for the estimator using orthogonalized version of Gaussian vectors) then whether the* MSE *is larger or smaller than for the standard unstructured case is determined by the sign of the sum $\sum_{i \neq j}\delta_{i,j}$. For the estimator using orthogonalized version of Gaussian vectors we have already showed that for every $i \neq j$ we have: $\delta_{i,j} > 0$ thus we obtain estimator with smaller* MSE. *If $\mathbf{M}$ is a product of blocks $\mathbf{HD}$ then we both have: an estimator with dependent rows and with bias. In that case it is also easy to see that $\mathbb{P}(\mathcal{A}^i)$ does not depend on the choice of $i$. Thus there exists some $\epsilon$ such that $\epsilon = \mathbb{P}(\mathcal{A}^i) - \frac{\theta}{\pi}$. Thus the estimator based on the $\mathbf{HD}$ blocks gives smaller* MSE *iff:*

$$\sum_{i \neq j}\delta_{i,j} + m\epsilon^2 < 0.$$

## 10   Further comparison of variants of OJLT based on SD-product matrices

In this section we give details of further experiments complementing the theoretical results of the main paper. In particular, we explore the various parameters associated with the **SD**-product matrices introduced in §2. In all cases, as in the experiments of §6, we take the structured matrix **S** to be the normalized Hadamard matrix **H**. All experiments presented in this section measure the MSE of the OJLT inner product estimator for two randomly selected data points in the g50c data set. The MSE figures are estimated on the basis of 1,000 repetitions. All results are displayed in Figure 3.

(a) Comparison of estimators based on **S**-Rademacher matrices with a varying number of **SD** matrix blocks, using the with replacement sub-sampling strategy.

(b) Comparison of estimators based on **S**-Rademacher matrices with a varying number of **SD** matrix blocks, using the sub-sampling strategy without replacement.

(c) Comparison of the use of $\mathbf{M}_{\mathbf{S}\mathcal{R}}^{(3)}$, $\mathbf{M}_{\mathbf{S}\mathcal{H}}^{(3)}$, and $\mathbf{M}_{\mathbf{S}\mathcal{U}}^{(3)}$ (introduced in §8.7) for dimensionality reduction. All use sub-sampling without replacement. The curves corresponding to the latter two random matrices are indistinguishable.

Figure 3: Results of experiments comparing OJLTs for a variety of **SD**-matrices.