[Reviews · NeurIPS 2017]

Reviewer 1



The paper examines embeddings based on structured matrices. In particular the paper analyzes the expected reconstruction error of a class of pointwise non-linear gaussian kernels computed using the embedded vectors. Embeddings based on structured matrices are well known in literature, [Sarlos, Smola '13, Yu et al '16] and have been studied from a practical and a theoretical viewpoint. In particular it is proven that they achieve an error, that is equal, up to constants to the one of unstructured matrices. The main contribution of this paper is to show that the constant is smaller than one ( it tends to 1 when the ambient dimension tend to infinity). The paper is technically correct. Note that the crucial aspect for which the structured methods are preferred with respect to the unstructured ones is that they require O(n log n) instead O(n^2) to be computed, while having a comparable accuracy with respect to the unstructured ones, as widely proven in literature. The proposed bound give a constant smaller than one. However the constants of the previous bounds comparing the error of structured and unstructured methods are already small and universal and the proposed bound does not reduce the error rate w.r.t. the ambient dimension or the number of random features. So the contribution consists in a minor improvement on the knowledge of the topic. ------- Reading the rebuttal didn't change my point of view on the paper. Again I remark that the paper provides a result that is of interest and I think it should be accepted. However the proposed result is more on the technical side and does not consist in a major improvement on the topic (e.g. compared to [Yu et al '16], which indeed received an oral presentation).

Reviewer 2



The paper analyses the theoretical properties of a family random projections approaches to approximate inner products in high dimensional spaces. In particular the authors focus on methods based on random structured matrices, namely Gaussian orthogonal matrices and SD-matrices. The latter are indeed appealing since they require significantly less computations to perform the projections thanks to their underlying structure. The authors show that the methods considered perform comparably well (or better) with respect to the Johnson-Lindenstrauss transform (baseline based on unstructured Gaussian matrices). Moreover they show that further improvements can be achieved by extending SD-matrices to the complex domain. The authors extend their analysis to the case random feature based approximation of angular kernels. The paper is well written and clear to read, however the discussion of some of the results could be elaborated more. For instance after Thm 3.3, which characterizes the MSE of the orthogonal JL transform based on SD-matrices, it is not discussed in much detail how this compares to the standard JL baseline. Cor. 3.4 does not really help much since it simply states that Thm 3.3 yields to lower MSE, without clarifying the entity of such improvement. In particular it appears that the improvement in performance of the OJLT over JLT is only in terms of constants (w.r.t. the number of sub-sampled dimensions m). I found it a bit misleading in Sec. 4, to introduce a general family of kernels, namely the pointwise nonlinear Gaussian kernels, but then immediately focus on a specific instance of such class. The reader expects the following results to apply to the whole family but this is not the case. Reversing the order, and discussing PNG kernels only at the end of the section would probably help the discussion. I found the term 'Unreasonable' in the title not explained in the text. Is it not reasonable to expect that adding structure to an estimator could make it more effective? The Mean Squared Error (MSE) is never defined. Although being a standard concept, it is also critical to the theoretical analysis presented in the paper, so it should be defined nevertheless.

Reviewer 3



This paper analyzes a few extensions of the random orthogonal embeddings proposed by Yu et al 2016. In particular, the authors focus on the Gaussian orthogonal matrices and SD-product matrices (which slightly generalize the HD-product matrices seen in the past). The authors are able to show several results regarding the unbiasedness and the MSE of the embeddings, for the linear kernel and the angular kernel (recall that the Yu et al paper performed analysis on the Gaussian kernel). One interesting result is that for SD-product matrices, the MSE for linear kernel exhibits osculation when the number of SD blocks alternate between odd and even. Such a result supports the empirical finding that taking 3 blocks works well. There are quite a few technical developments in the paper (30 pages of appendix) and it is impossible to verify every piece given the short turnaround. I read some proofs and they are correct. I would suggest giving a concrete mathematical definition of the Gaussian orthogonal matrix in Section 2.